# Assessing the degree of detail of temperature-based snow routines for runoff modelling in mountainous areas in Central Europe

Marc Girons Lopez[1,2], Marc J. P. Vis[1], Michal Jenicek[3], Nena Griessinger[4], Jan Seibert[1,5]

[1]Department of Geography, University of Zurich, Zurich, CH-8006, Switzerland
[2]Swedish Meteorological and Hydrological Institute, Norrköping, SE-60176, Sweden
[3]Department of Physical Geography and Geoecology, Charles University, Prague, CZ-12843, Czechia
[4]WSL Institute for Snow and Avalanche Research SLF, Davos, CH-7260, Switzerland
[5]Department of Aquatic Sciences and Assessment, Swedish University of Agricultural Sciences, Uppsala, SE-75007 Sweden

*Correspondence to*: Marc Girons Lopez (marc.girons@smhi.se)

**Abstract.** Snow processes are a key component of the water cycle in mountainous areas as well as in many areas of the mid- and high latitudes of the Earth. The complexity of these processes, coupled with the limited data available on them, has led to the development of different modelling approaches aimed at improving our understanding of these processes and supporting decision-making and management practices. Physically-based approaches, such as the energy balance method, provide the best representation of snow processes but limitations in data availability in many situations constrain their applicability in

favour of more straightforward approaches. Indeed, the comparatively simple temperature-index method has become the most widely-used modelling approach for representing snowpack processes in rainfall-runoff modelling, with different variants of this method implemented across many models. Nevertheless, the decisions on the most suitable degree of detail of the model are in many cases not adequately assessed for a given application.

In this study we assessed the suitability of a number of formulations of different components of the simple temperature-index

method for rainfall-runoff modelling in mountainous areas of Central Europe by using the HBV bucket-type model. To this end, we reviewed the most widely-used formulations of different components of temperature-based snow routines from different rainfall-runoff models and proposed a series of modifications to the default structure of the HBV model. We narrowed the choice of alternative formulations to those that provide a simple conceptualisation of the described processes in order to constrain parameter and model uncertainty. We analysed a total of 64 alternative snow routine structures over 54 catchments

using a split-sample test. Overall, the most valuable modifications to the standard structure of the HBV snow routine were (a) using an exponential snowmelt function coupled with no refreezing, and (b) computing melt rates with a seasonally-variable degree-day factor. Our results also demonstrated that increasing the degree of detail of the temperature-based snow routines in rainfall-runoff models did not necessarily lead to an improved model performance per se. Instead, performing an analysis on which processes are to be included, and to which degree of detail, for a given model and application is a better approach to

obtain more reliable and robust results.

# 1 Introduction

Snow is an essential aspect of the annual hydrological variations in Alpine areas as well as in many other regions of the mid and high latitudes of the Earth. Unlike rainfall, which contributes directly to the groundwater recharge and stream runoff, snowfall accumulates on the ground creating a temporary freshwater reservoir. This accumulated water is then gradually released through melting when the necessary energy for melt is available, contributing to runoff. Incoming solar radiation is the major control of the variability of the available energy whilst air temperature is a good proxy for the variation of the available energy and, thus, snowmelt (Sicart et al., 2008). The snow accumulated on the ground (i.e., snowpack) is not only crucial for ecological reasons (Hannah et al., 2007), but also for many human activities such as hydropower, agriculture, or tourism (Barnett et al., 2005). At the same time, snow processes can also lead to risks for society. For instance, the accumulation of snow on steep slopes may cause avalanches (Schweizer et al., 2003), and the sudden melt of large amounts of snow during rain-on-snow events (Sui and Koehler, 2001) or after a rapid increase of air temperature, may lead to widespread flooding (Merz and Blöschl, 2003; Rico et al., 2008).

Society´s dependence on the freshwater stored in the snowpack and its vulnerability to its associated risks raises the need to understand its dynamics and evolution (Fang et al., 2014; Jamieson and Stethem, 2002). Nevertheless, while knowledge on snow hydrology has broadly advanced over the last decades with, for instance, the establishment of experimental catchments devoted to snow processes research (Pomeroy and Marks, 2020), or the use of remote sensing data for snowmelt monitoring (Dietz et al., 2012), limited observations in most locations still pose a challenge to properly quantifying snow processes and implementing adequate management policies and practices. In addition to present-day limitations, the evolution of snow water resources in the future, which cannot be estimated through direct observations, is also essential in the context of global climate change (Berghuijs et al., 2014; Jenicek and Ledvinka, 2020).

Different modelling strategies have been developed to overcome the data limitations and to study the evolution of the snowpack and its impact on water resources. The most common modelling approaches are based either on the physically-based energy budget model or on the temperature-index approach (Verdhen et al., 2014). While energy budget models are the most accurate alternative to represent snowpack processes, they usually require data that are often not available at conventional meteorological stations (Avanzi et al., 2016). These models attempt to estimate the snow contribution to runoff, generally in a distributed way, by solving the energy balance of the snowpack, which requires detailed data on topography, temperature, wind speed and direction, cloud cover fraction, snow density, etc. Some efforts have also been done to implement such approaches at sub-catchment or even catchment scales, thus requiring less driving data (Skaugen et al., 2018). Temperature-based models (also known as temperature-index or degree-day methods), in contrast, are based on the assumption that the temporal variability of incoming solar radiation is well represented by the variations of air temperature (Ohmura, 2001; Sicart et al., 2008) and tend to have low, and thus easy to meet data requirements and computational demands, and offer a satisfactory balance between simplicity and performance, which makes them successful in many different contexts and applications, even in cases with limited data availability (Hock, 2003). Nevertheless, the assumption that incoming solar radiation is well

represented by air temperature does not always hold, such as in high elevation catchments where temperature seldom raises above the freezing point (Gabbi et al., 2014; Pellicciotti et al., 2005), or in conditions in which sublimation from the snowpack becomes a significant process (Herrero and Polo, 2016). Such issues led to the development of extended formulations including additional variables such as wind speed or relative humidity to improve the snowmelt estimation (Zuzel and Cox, 1975) or even hybrid methods combining energy-based and temperature-based approaches, such as the inclusion of a radiation component in temperature-based models (Hock, 1999; Kane et al., 1997).

Simple temperature-index models define the rate of snowmelt as being proportional to the temperature above the freezing point per unit time through a proportionality constant commonly named degree-day factor (Collins, 1934; Martinec, 1960). Many conceptual rainfall-runoff models use variations of this method to simulate snowpack processes. For instance, while many models use a simple formulation including a constant degree-day factor both in time and space (Valéry et al., 2014), others include a monthly or seasonally variable parameter (Hottelet et al., 1994; Quick and Pipes, 1977) or even a spatially variable degree-day factor that takes, amongst others, differences in slope, aspect, or vegetation cover into account (He et al., 2014). Additionally, while some models use the freezing point (i.e. 0°C) as the threshold temperature for the onset of snowmelt (Walter et al., 2005), others include a calibrated parameter(Viviroli et al., 2007) to allow for spatial variations on this process. Furthermore, some models disregard some of the processes, such as refreezing, as their magnitude tends to be negligible with respect to snowmelt (Magnusson et al., 2014). Other components of the snow routine may also be conceptualised with different degrees of detail. A good example is the formulation of the precipitation phase partition between rain and snow. While some models set a sharp threshold for this transition, others use a gradual transition where rain and snow may occur at the same time, using different model formulations and, in some cases, also additional data such as relative humidity (Matsuo and Sasyo, 1981) to define this transition. In general, however, the inherent simplifications made in semi-distributed temperature-index models leave out some critical aspects of the snowpack processes that may be significant in some circumstances. For instance, the disregarding of lateral transport processes in many models may lead to the development of unreasonable accumulations of snow over long periods (i.e., snow towers) in high mountainous areas (Freudiger et al., 2017; Frey and Holzmann, 2015).

Overall, the degree of detail in which the different snow processes are formulated in different models differs greatly, and depends to a great extent on the model philosophy and preferences, purpose, application, desired resolution, or available data and computing power, among others. Nevertheless, these choices are not always adequately taken into account when using a specific model for a different application or purpose to what it was originally developed for (Harpold et al., 2017), may lead to models with a more detailed representation of hydrological processes performing worse than comparatively more simplistic models for a specific purpose (Orth et al., 2015). So, models (or the relevant model routines) should always be tested beforehand to ensure that the assumptions and formulations used are adequate and robust for the intended application (Günther et al., 2019). For a long time, however, limitations in computing power hindered the systematic testing of different model structures over a large number of catchments. In recent years, however, the increase in computing power has made these tests not only feasible but also desirable.

In this study, we present a methodology to evaluate the design choices of a rainfall-runoff model with a simple temperature-based snow routine for its application over a large number of catchments. More specifically, we aim to evaluate the suitability of the snow routine of the HBV rainfall-runoff model (Bergström, 1995), a typical bucket-type model, for its application in mountainous catchments in Central Europe. Taking the existing model structure as a reference, we implemented and tested a number of model structure modifications based on common formulations of snow processes in other rainfall-runoff models with simple temperature-based snow routines to assess the most suitable model structure for the intended application. That is, model structures which generally result in improved model performance for representing both snow processes and stream runoff in the area of interest while avoiding adding unnecessary elements and parameters that would result in increased model uncertainty and equifinality issues (Beven, 2008). To ensure that the results are representative, we explored different levels of added detail, from modifications to single components of the snow routine to combinations of modifications to multiple components on a large dataset of catchments covering a wide range of geographical, climatological, and hydrological conditions of the area of interest.

## 2 Materials and methods

The HBV model is a bucket-type rainfall-runoff model with a number of routines representing the main components of the terrestrial part of the water cycle, i.e., snow routine, soil routine, groundwater (response) routine, and routing function. In this study, we focused solely on the snow routine of the model. We used the HBV-light software, which follows the general structure of other implementations of the HBV model and includes some additional functionalities such as Monte Carlo runs and a genetic algorithm for automated optimisation (Seibert and Vis, 2012). Henceforth we use the term 'HBV model' when referring to our simulations using the default HBV-light software. That is, the HBV model with the snow routine as described in Lindström et al. (1997) and Seibert & Vis (2012).

### 2.1 HBV's snow routine

The snow routine of the HBV model is based on widely-used and well-tested conceptualisations of the relevant snow processes for rainfall-runoff modelling. More specifically, it represents the main processes related to (i) the precipitation phase partition between snow and rain, and (ii) the snow accumulation and subsequent melt and refreezing cycles of the snowpack.

Regarding the precipitation phase partition, HBV uses a threshold temperature parameter, $T_T$ $[°C]$, above which all precipitation, $P$ $[mm\ \Delta t^{-1}]$, is considered to fall as rain, $P_R$ $[mm\ \Delta t^{-1}]$ (Eq. 1). This threshold can be adjusted to account for local conditions. Below the threshold, all snow is considered to fall as snow, $P_S$ $[mm\ \Delta t^{-1}]$ (Eq. 2). The combined effect of snowfall undercatch and interception of snowfall by the vegetation is represented by a snowfall correction factor, $C_{SF}$ $[-]$.

$$P_R = P, \quad \&T > T_T \ , \tag{1}$$

$$P_S = P \cdot C_{SF}, \quad \&T \leq T_T \ , \tag{2}$$

As previously mentioned, the HBV model uses a simple approach based on the temperature-index method to simulate the evolution of the snowpack. This way, snowmelt, $M$ *[mm $\Delta t^{-1}$]*, is assumed to be proportional to the air temperature, $T$ *[°C]*, above a predefined threshold temperature, $T_T$ *[°C]*, through a proportionality coefficient, also called degree-day factor, $C_0$ *[mm $\Delta t^{-1}$ °$C^{-1}$]* (Eq. 3). The model allows for a certain volume of melted water to remain within the snowpack, given as a fraction of the corresponding snow water equivalent of the snowpack, $C_{WH}$ *[-]*. Finally, refreezing of melted water, $F$ *[mm $\Delta t^{-1}$]* takes place when the air temperature is below $T_T$, and its magnitude is modulated through an additional proportionality parameter, $C_F$ *[-]* (Eq. 4).

$$M = C_0(T - T_T) \,, \tag{3}$$

$$F = C_F \cdot C_0(T_T - T) \,, \tag{4}$$

Overall, the snow routine of the HBV model contains five calibration parameters. HBV allows for a limited representation of catchment characteristics through the specification of different elevation and vegetation zones. This way, the parameters controlling the different processes included in the snow routine can be modified for individual vegetation zones. The combination of elevation and vegetation zones (also known as Elevation Vegetation Units, EVUs) is the equivalent of the Hydrologic Response Units (HRUs) used in other conceptual models (Flügel, 1995). Both precipitation and temperature are corrected for elevation using respective lapse-rate parameters.

## 2.2 Proposed modifications to individual components of the snow routine

Here we review the different components of the snow routine structure of the HBV model as well as functions that are directly related to this routine (e.g. input data correction with elevation) and describe the proposed modifications to each component. Each of these alternative structures requires one to three additional parameters (Table 1).

**Table 1 Description of the proposed modifications to the snow routine of the HBV model. The default component structures of the HBV model are marked with a * symbol. The components marked with a † are not formally part of the snow routine but were included in the analysis due to their significant impact on it.**

| Snow routine component | Structure | Abbreviation | Number of additional parameters |
| --- | --- | --- | --- |
| Precipitation lapse rate† | Constant* | - | 1 |
| Temperature lapse rate† | Constant* | $\Gamma_c$ | 1 |
| | Seasonally-variable | $\Gamma_s$ | 2 |
| Precipitation phase partition | Abrupt transition* | $\Delta P_a$ | 1 |
| | Partition defined by a linear function | $\Delta P_l$ | 2 |
| | Partition defined by a sine function | $\Delta P_s$ | 2 |
| | Partition defined by an exponential function | $\Delta P_e$ | 2 |

| | | | |
|---|---|---|---|
| Threshold temperature | One threshold for both precipitation and snowmelt* | $T_T$ | 1 |
| | Different thresholds for precipitation and snowmelt | $T_{P,M}$ | 2 |
| Degree-day factor | Constant* | $C_{0,c}$ | 1 |
| | Seasonally-variable | $C_{0,s}$ | 2 |
| Snowmelt and refreezing | Linear snowmelt and refreezing magnitude increase with temperature* | $M_l$ | 3 |
| | Exponential snowmelt magnitude increase with temperature. No refreezing. | $M_e$ | 3 |

### 2.2.1 Temperature and precipitation lapse rates

When different elevation zones are used, the temperature for each zone is generally computed from some catchment-average value and a lapse rate parameter. In HBV, a constant temperature lapse rate is usually used. Alternatively, if the available data allows, it is also possible to provide an estimation of the daily temperature lapse rate. However, if no data on the altitude
dependence of temperature is available, setting a constant value throughout the year might be an oversimplification. Indeed, in an experimental study on several locations across the Alps, Rolland (2002) found that the seasonal variability of the temperature lapse rate follows approximately a sine curve with a minimum around the winter solstice. Following these findings, we implemented a seasonally variable computation of the temperature lapse rate using a sine function (Eq. 5). This way, the temperature lapse rate for a given day of the year, $\Gamma_n$ [°C 100 m$^{-1}$] (where $n$ is the day of the year, a sequential day number
starting with 1 on the 1$^{st}$ of January), depends on two parameters, namely the annual temperature lapse rate average, $\Gamma_0$ [°C 100 m$^{-1}$], and amplitude, $\Gamma_i$ [°C 100 m$^{-1}$].

$$\Gamma_n = \Gamma_0 + \frac{1}{2}\Gamma_i \sin\frac{2\pi(n-81)}{365}, \tag{5}$$

Precipitation lapse rates cannot be related to seasonal or other types of systematic variations as they are strongly dependent on the synoptic meteorological conditions and therefore highly variable. Consequently, we decided to keep the default approach
in the HBV model which consists in calibrating the model using a constant precipitation lapse rate parameter.

### 2.2.2 Precipitation phase partition

The determination of the precipitation phase is a crucial step as it controls whether water accumulates in the snowpack or contributes directly to recharge and runoff. In the HBV model, the distinction between rainfall and snowfall is based on the assumption that precipitation falls either as rain or as snow, depending on a threshold temperature parameter. However, in
reality, this transition is less sharp, as both rain and snow can coincide (Dai, 2008; Magnusson et al., 2014; Sims and Liu, 2015). Additionally, depending on other factors such as humidity and atmosphere stratification, the shift from rain to snow can

occur at different temperatures. Therefore, the single threshold temperature may not adequately represent the snow accumulation, especially in areas or periods with temperatures close to zero degrees Celsius. Different formulations have been proposed to describe the snow fraction of precipitation, $S$ $[-]$, as a function of temperature (Froidurot et al., 2014; Magnusson et al., 2014; Viviroli et al., 2007). In this study, we considered three different formulations to calculate the snowfall fraction of precipitation (Eq. 6 and 7, respectively): (i) a linear function (Eq. 8), (ii) a sine function (Eq. 9), and (iii) an exponential function (Eq. 10). Both the $T_A$ $[°C]$ and $M_P$ $[°C]$ parameters control the range of temperatures for mixed precipitation.

$$P_S = P \cdot S \cdot C_{SF} \,, \tag{6}$$

$$P_R = P \cdot (1 - S) \,, \tag{7}$$

$$S = \begin{cases} 1, & T \le T_T - \frac{T_A}{2} \\ \frac{1}{2} + \frac{T_T - T}{T_A}, & T_T - \frac{T_A}{2} < T \le T_T + \frac{T_A}{2}, \\ 0, & T > T_T + \frac{T_A}{2} \end{cases} \tag{8}$$

$$S = \begin{cases} 1, & T \le T_T - \frac{T_A}{2} \\ \frac{1}{2} - \frac{1}{2}\sin\left(\pi\frac{T_T - T}{T_A}\right), & T_T - \frac{T_A}{2} < T \le T_T + \frac{T_A}{2}, \\ 0, & T > T_T + \frac{T_A}{2} \end{cases} \tag{9}$$

$$S = \frac{1}{1 + e^{\frac{T - T_T}{M_P}}} \,, \tag{10}$$

### 2.2.3 Snowmelt threshold temperature

In addition to determining the precipitation phase, a temperature threshold parameter is also needed to determine the onset of snowmelt. The most straightforward approach, used in the HBV model, is to use the same threshold temperature parameter for both snowfall and snowmelt. However, as these two transitions are related to different processes happening at different environmental conditions, a single parameter might not adequately describe both transitions. A more realistic approach would be to consider two separate parameters for these processes: a threshold temperature parameter for precipitation phase partitioning, $T_P$ $[°C]$, and another one for snowmelt and refreezing processes, $T_M$ $[°C]$ (Debele et al., 2010).

### 2.2.4 Degree-day factor

The degree-day factor is an empirical factor that relates the rate of snowmelt to air temperature (Ohmura, 2001). In the HBV model, a simple proportionality coefficient to estimate the magnitude of the snowmelt is used. This coefficient, multiplied by a constant (usually set to 0.05 in HBV), is also used to compute refreezing rates. Nevertheless, while the degree-day factor is often assumed to be constant over time, seasonal changes in snow albedo and solar inclination point out to temporal variations of the degree-day factor as well. While some models use monthly values for this parameter (Quick and Pipes, 1977), a more

elegant but still simple way to represent this variability is to consider a seasonally variable degree-day factor following a sine function defined by a yearly average degree-day factor parameter, $C_0$ *[mm $\Delta t^{-1}$ °$C^{-1}$]*, and an amplitude parameter, $C_{0,a}$ *[mm $\Delta t^{-1}$ °$C^{-1}$]*, defining the amplitude of the seasonal variation (Eq. 11) (Braun and Renner, 1992; Hottelet et al., 1994). By establishing a seasonally-variable degree-day factor instead of a constant value for this parameter, potential snowmelt rates become smaller during the winter months, and increase during spring and summer (if there is any snow left).

$$C_{0,n} = C_0 + \frac{1}{2} C_{0,a} \sin \frac{2\pi(n-81)}{365} \, , \tag{11}$$

### 2.2.5 Snowmelt and refreezing

All liquid water produced by snowmelt does not leave the snowpack directly, as a certain amount of liquid water can be stored in the snow thus delaying the outflow of water from the snowpack. The liquid water stored in the snowpack can also refreeze if temperatures decrease below the freezing point. In the HBV model, both the storage of liquid water and refreezing processes are considered. However, since the magnitude of refreezing meltwater is generally tiny compared to other fluxes, some models disregard this process entirely to reduce model complexity (Magnusson et al., 2014). Here we follow the approach by Magnusson et al. (2014) which, besides disregarding the refreezing process, describes the snowmelt magnitude using an exponential function (Eq. 12). This formulation of snowmelt is somewhat more detailed than the one used in HBV and requires the use of an additional parameter to control for the smoothness of the snowmelt transition, $M_M$ *[°C]*. Contrary to the formulation used in the standard HBV model, snowmelt occurs even below the freezing point, but at negligible amounts. The impact of increasing temperature on snowmelt is higher for this formulation compared to HBV.

$$M = C_0 \cdot M_M \left[ \frac{T-T_T}{M_M} + \ln \left( 1 + e^{-\frac{T-T_T}{M_M}} \right) \right] , \tag{12}$$

### 2.3 Study domain and data

We selected two sets of mountainous catchments located at different countries within Central Europe to test the proposed modifications to the individual components of the snow routine of the HBV model (Table 2, Figure 1). The first set, composed of Swiss catchments, contains catchments ranging from high-altitude, steep catchments in the central Alps to low-altitude catchments in the Pre-Alps and Jura mountains with gentler topography. The second set, composed of Czech catchments, is representative of mountain catchments at lower elevations compared to the Swiss catchments.

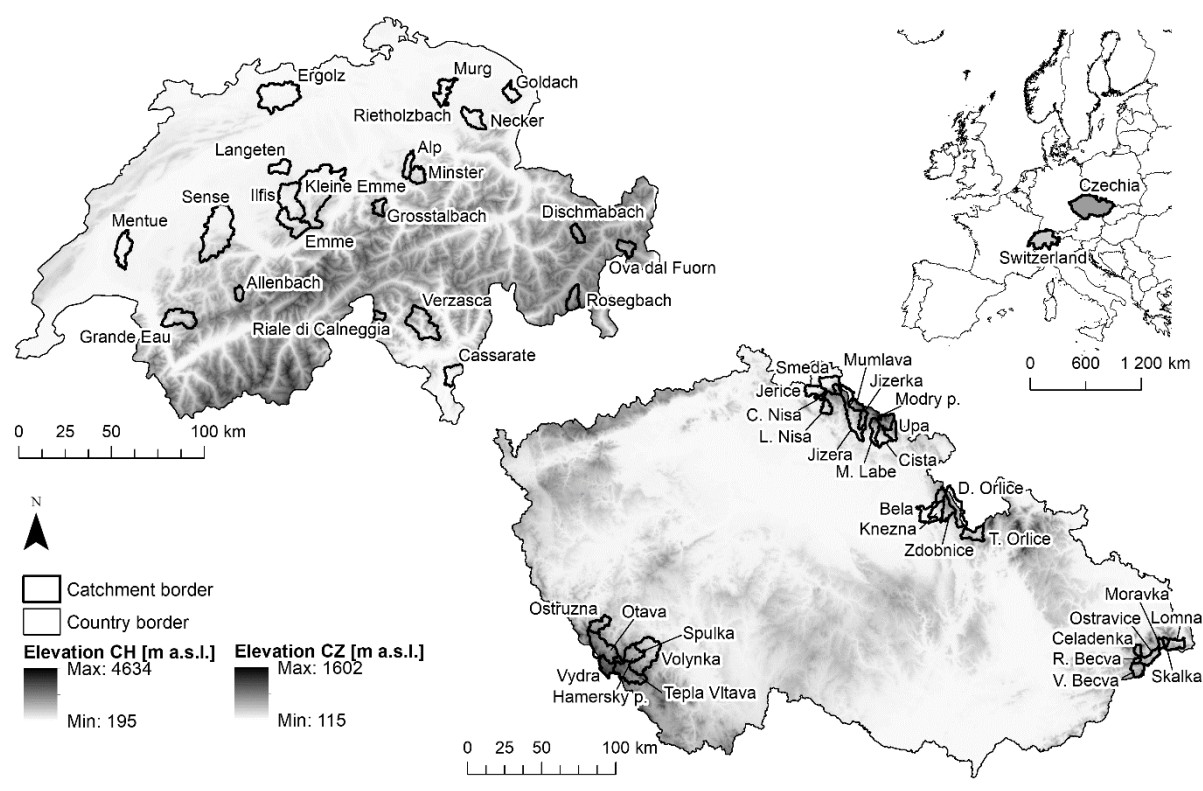

**Figure 1 Geographical location of the catchments used in this study. We used a total of 54 catchments; 22 located in Switzerland and 32 in Czechia.**

**Table 2 Relevant physical characteristics of the catchments included in the study. Each catchment is given an identification code in**
**the following way: country (CH – Switzerland, CZ – Czechia), geographical location (Switzerland: 100 – Jura and Swiss Plateau, 200 – Central Alps, 300 – Southern Alps; Czechia: 100 – Bohemian Forest, 200 – Western Sudetes, 300 – Central Sudetes, 400 – Carpathians), and a sequential number for increasingly snow-dominated catchments within each geographical setting. The official hydrometric station IDs from FOEN and CHMI are also provided.**

| ID | Catchment | Station | Station ID | Area [km²] | Mean elevation [m a.s.l.] | Elevation range [m a.s.l.] | Snowmelt contribution to runoff [%] |
|----|-----------|---------|------------|------------|---------------------------|----------------------------|-------------------------------------|
| CH-101 | Ergolz | Liestal | 2202 | 261.2 | 604 | 305 – 1087 | 5 |
| CH-102 | Mentue | Yvonand | 2369 | 105.3 | 690 | 469 – 915 | 5 |
| CH-103 | Murg | Wängi | 2126 | 80.1 | 657 | 469 – 930 | 7 |
| CH-104 | Langeten | Huttwil | 2343 | 59.9 | 770 | 632 – 1032 | 8 |
| CH-105 | Goldach | Goldach | 2308 | 50.4 | 825 | 401 – 1178 | 14 |
| CH-106 | Rietholzbach | Mosnang | 2414 | 3.2 | 774 | 697 – 868 | 9 |

| | | | | | | | |
|---|---|---|---|---|---|---|---|
| CH-107 | Sense | Thörishaus | 2179 | 351.2 | 1091 | 551 – 2096 | 12 |
| CH-108 | Emme | Eggiwil | 2409 | 124.4 | 1308 | 770 – 2022 | 22 |
| CH-109 | Ilfis | Langnau | 2603 | 187.4 | 1060 | 699 – 1973 | 14 |
| CH-110 | Alp | Einsiedeln | 2609 | 46.7 | 1173 | 878 – 1577 | 19 |
| CH-111 | Kleine Emme | Emmen | 2634 | 478.3 | 1080 | 440 – 2261 | 16 |
| CH-112 | Necker | Mogelsberg | 2374 | 88.1 | 970 | 649 – 1372 | 16 |
| CH-113 | Minster | Euthal | 2300 | 59.1 | 1362 | 891 – 1994 | 26 |
| CH-201 | Grande Eau | Aigle | 2203 | 131.6 | 1624 | 427 – 3154 | 26 |
| CH-202 | Ova dal Fuorn | Zernez | 2304 | 55.2 | 2359 | 1797 – 3032 | 36 |
| CH-203 | Grosstalbach | Isenthal | 2276 | 43.9 | 1880 | 781 – 2700 | 28 |
| CH-204 | Allenbach | Adelboden | 2232 | 28.8 | 1930 | 1321 – 2587 | 38 |
| CH-205 | Dischmabach | Davos | 2327 | 42.9 | 2434 | 1657 – 3024 | 52 |
| CH-206 | Rosegbach | Pontresina | 2256 | 66.6 | 2772 | 1771 – 3793 | 62 |
| CH-301 | Riale di Calneggia | Cavergno | 2356 | 23.9 | 2079 | 881 – 2827 | 42 |
| CH-302 | Verzasca | Lavertezzo | 2605 | 185.1 | 1723 | 546 – 2679 | 27 |
| CH-303 | Cassarate | Pregassona | 2321 | 75.8 | 1017 | 286 – 1904 | 4 |
| CZ-101 | Vydra | Modrava | 135000 | 89.8 | 1140 | 983 – 1345 | 34 |
| CZ-102 | Otava | Rejstejn | 137000 | 333.6 | 1017 | 598 – 1345 | 29 |
| CZ-103 | Hamersky potok | Antygl | 136000 | 20.4 | 1098 | 978 – 1213 | 26 |
| CZ-104 | Ostruzna | Kolinec | 139000 | 92.0 | 755 | 541 – 1165 | 17 |
| CZ-105 | Spulka | Bohumilice | 141700 | 104.6 | 804 | 558 – 1131 | 19 |
| CZ-106 | Volynka | Nemetice | 143000 | 383.4 | 722 | 430 – 1302 | 17 |
| CZ-107 | Tepla Vltava | Lenora | 106000 | 176.0 | 1010 | 765 – 1314 | 20 |
| CZ-201 | Jerice | Chrastava | 319000 | 76.0 | 493 | 295 – 862 | 14 |
| CZ-202 | Cerna Nisa | Straz nad Nisou | 317000 | 18.3 | 672 | 368 – 850 | 13 |
| CZ-203 | Luzicka Nisa | Prosec | 314000 | 53.8 | 611 | 419 – 835 | 22 |
| CZ-204 | Smeda | Bily potok | 322000 | 26.5 | 817 | 412 – 1090 | 26 |
| CZ-205 | Smeda | Frydlant | 323000 | 132.7 | 588 | 297 – 1113 | 18 |
| CZ-206 | Jizera | Dolni Sytová | 086000 | 321.8 | 771 | 399 – 1404 | 26 |
| CZ-207 | Mumlava | Janov-Harrachov | 083000 | 51.3 | 970 | 625 – 1404 | 34 |
| CZ-208 | Jizerka | Dolni Stepanice | 086000 | 44.2 | 842 | 490 – 1379 | 29 |
| CZ-209 | Malé Labe | Prosecne | 003000 | 72.8 | 731 | 376 – 1378 | 25 |
| CZ-210 | Cista | Hostinne | 004000 | 77.4 | 594 | 358 – 1322 | 19 |
| CZ-211 | Modry potok | Modry dul | 008000 | 2.6 | 1297 | 1076 – 1489 | 38 |
| CZ-212 | Upa | Horni Marsov | 013000 | 82.0 | 1030 | 581 – 1495 | 28 |
| CZ-213 | Upa | Horni Stare Mesto | 014000 | 144.8 | 902 | 452 – 1495 | 25 |
| CZ-301 | Bela | Castolovice | 031000 | 214.1 | 491 | 269 – 1104 | 25 |

| CZ-302 | Knezna | Rychnov nad Kneznou | 030000 | 75.4 | 502 | 305 – 861 | 25 |
| CZ-303 | Zdobnice | Slatina nad Zdobnici | 027000 | 84.1 | 721 | 395 – 1092 | 24 |
| CZ-304 | Divoka Orlice | Klasterec nad Orlici | 024000 | 153.6 | 728 | 505 – 1078 | 22 |
| CZ-305 | Ticha Orlice | Sobkovice | 032000 | 98.5 | 622 | 459 – 965 | 22 |
| CZ-401 | Vsetinska Becva | Velke Karlovice | 370000 | 68.3 | 749 | 524 – 1042 | 22 |
| CZ-402 | Roznovska Becva | Horni Becva | 383000 | 14.1 | 745 | 568 – 966 | 24 |
| CZ-403 | Celadenka | Celadna | 279000 | 31.0 | 803 | 536 – 1187 | 30 |
| CZ-404 | Ostravice | Stare Hamry | 275300 | 73.3 | 707 | 542 – 922 | 32 |
| CZ-405 | Moravka | Uspolka | 281000 | 22.2 | 763 | 560 – 1104 | 30 |
| CZ-406 | Skalka | Uspolka | 282000 | 18.9 | 785 | 571 – 1029 | 24 |
| CZ-407 | Lomna | Jablunkov | 298000 | 69.9 | 667 | 390 – 1011 | 25 |

### 2.3.1 Switzerland

We selected 22 catchments in Switzerland covering a wide range of elevations and areas in the three main hydro-geographical domains of the country, i.e. the Jura and Swiss Plateau, the Central Alps, and the Southern Alps (Weingartner and Aschwanden, 1989). No catchments with significant karst or glacierised areas, as well as catchments with substantial human influence on runoff were selected for this study. This decision allowed us to observe the signal of snow processes, without including noise or added complexity from other processes, but limited the number of catchments in high altitudes, which are the ones with largest snowmelt contribution to runoff. The resulting set of catchments had mean elevations between 600 and 2800 m a.s.l. with elevation gradients of up to 2000 m and catchment areas between 3 and 500 km$^2$ (Figure 2). There was a considerable variability in the yearly snowmelt contribution to runoff, ranging from 5 % to 60 % as the catchments ranged from pluvial to glacio-nival regimes.

We obtained the necessary meteorological data for running the HBV model from the Swiss Federal Office of Meteorology and Climatology (MeteoSwiss). More specifically, we used pre-processed gridded data products to obtain catchment-average precipitation (Frei et al., 2006; Frei and Schär, 1998), and temperature (Frei, 2014). These gridded data products are available from 1961, have a daily temporal resolution, and a spatial resolution of 1.25 degree minutes covering the entire country.

We used both stream runoff and snow water equivalent data for model calibration and validation. We obtained daily stream runoff data from the Swiss Federal Office for the Environment (FOEN, 2017). Regarding snow water equivalent, we used 18 years of gridded daily snow water equivalent data at 1 km$^2$ resolution derived from a temperature-index snow model with integrated three-dimensional sequential assimilation of observed snow data from 338 stations of the snow monitoring networks of MeteoSwiss and the Swiss Institute for Snow and Avalanche Research (SLF) (Griessinger et al., 2016; Magnusson et al., 2014). Even if using a temperature-index model for both the HBV model and the estimation of the snow water equivalent validation data may introduce some bias to the results, the data assimilation and error correction methods used in the estimation of snow water equivalent make this methodology especially robust (Magnusson et al., 2014). Finally, we obtained the

catchment areas and topography from a digital elevation model with a resolution of 25 m from the Swiss Federal Office of Topography (swisstopo, https://swisstopo.admin.ch/).

### 2.3.2 Czechia

The second set of catchments was composed of Czech catchments and includes 32 mountain catchments with catchment areas ranging from 3 to 383 km$^2$ (Figure 2). As for Switzerland, we selected near-natural catchments with no major human influences such as big dams or water transfers. The selected catchments were located at lower elevationsthan most of the selected Swiss catchments. Additionally, they were located in the transient zone between oceanic and continental climate, with lower mean annual precipitation than the Swiss catchments. The mean annual snow water equivalent peak for the period 1980 – 2014

ranged from 35 mm to 742 mm depending on catchment elevation, resulting in 13 % to 39 % of the annual runoff coming from spring snowmelt.

We obtained daily precipitation, daily mean air temperature, and daily mean runoff time series from the Czech Hydrometeorological Institute (CHMI). Additionally, we obtained weekly snow water equivalent data from CHMI (measured each Monday at 7 CET). Since no gridded precipitation, air temperature, or snow water equivalent data are available for

Czechia, station data were used for HBV model parametrization. We used stations located within the individual catchments when available. If no such station was available, we selected the nearest station representing similar conditions to the target catchment (e.g., stations situated at a similar elevation). Finally, we used a digital elevation model with a vertical resolution of 5 m from the Czech Office for Surveying, Mapping and Cadastre to obtain catchment areas and elevation distributions.

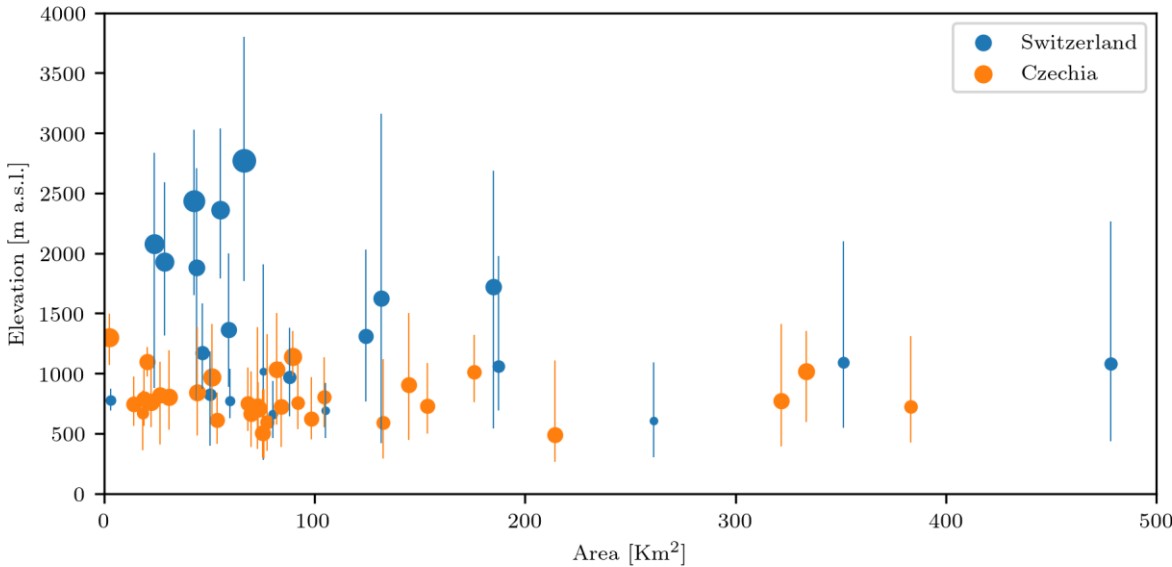

Figure 2 Distribution of the catchments used in this study in terms of area (x-axis), mean elevation and range (y-axis), and relative snowmelt contribution to runoff (marker size). The catchments are coloured according to their respective geographical domain: blue (Switzerland), and orange (Czechia).

### 2.4 Experimental setup

Even if sub-daily data were available for most variables for the Swiss catchments, we considered that daily data was beneficial for this study, as using sub-daily temporal resolutions would have required to take into account the diurnal variability of some of the variables, thus requiring a higher comprehensiveness over the included hydrological processes in the model (Wever et al., 2014). For instance, radiation and temperature fluctuations along the day would require similarly variable degree-day factor values (Hock, 2005). Other factors such as the transport time of meltwater from the snowpack to the streams would also become relevant at sub-daily time scales (Magnusson et al., 2015). To keep the model simple but at the same time being able to represent the elevation-dependent snow processes, we used a single vegetation zone per catchment but divided the catchment area into 100 m elevation zones (Uhlenbrook et al., 1999).

When evaluating the performance of rainfall-runoff models to simulate snow dynamics, this evaluation is sometimes done solely against runoff observations, as this variable is the main output of such models (Riboust et al., 2019; Watson and Putz, 2014). Nevertheless, this analysis alone is incomplete as the performance of the model to reproduce runoff is the result of the interaction between the different routines and components of the model, also those that are not directly related to snow processes. A direct evaluation of the relevant model routine (i.e. the snow routine in this case) should be performed as well. Focusing on the snow routine, snow cover fraction and snow water equivalent are widely-adopted evaluation metrics (Avanzi et al., 2016; Helbig et al., 2015). The fact that snow water equivalent is a more direct measure of the amount of water that will eventually be converted to runoff, in addition to the difficulties in accurately determining the snow cover fraction for our study

area and period, led us to choose snow water equivalent for evaluating the snow routine structure of the model. In short, we evaluated the different model structures based on their ability to represent (i) the snow water equivalent of the snowpack, and (ii) stream runoff at the catchment outlet.

To evaluate the performance of the different model structures to reproduce the snow water equivalent of the snowpack, we used a modified version of the Nash-Sutcliffe efficiency (Nash and Sutcliffe, 1970) where the model performance, $R_W$, is given by the fraction of the sum of quadratic differences between snow water equivalent observations, $W_O$, and simulations, $W_S$, and between observations and the mean observed value, $\overline{W_o}$ (Eq. 13).

$$R_W = 1 - \frac{\sum(W_o - W_s)^2}{\sum(W_o - \overline{W_o})^2} , \tag{13}$$

Due to the substantial differences data availability regarding in snow water equivalent (SWE) values between the two datasets (gridded data in Switzerland vs point data in Czechia), we had to adapt the model calibration and evaluation procedure to each case. We evaluated the model against the mean snow water equivalent value for each elevation zone for the Swiss catchments, and against the measured values at a given elevation for the Czech ones.

Regarding the evaluation of the model against stream runoff, we deemed that the standard Nash-Sutcliffe efficiency measure was not suitable for our case study as it is skewed towards high flows (Schaefli and Gupta, 2007). Snow processes are dominant both in periods of high flows (e.g. spring flood) and low flows (e.g. winter conditions), which are equally important for our purposes. For this reason, we decided to evaluate the estimation of stream runoff by using the natural logarithm of runoff instead (Eq. 14).

$$R_{\ln Q} = 1 - \frac{\sum(\ln Q_o - \ln Q_s)^2}{\sum(\ln Q_o - \overline{\ln Q_o})^2} , \tag{14}$$

Some studies focusing on snow hydrology establish specific calibration periods for each catchment based on, for instance, the snowmelt season (Griessinger et al., 2016). In this study, however, we decided to constrain the calibration and evaluation periods in a consistent and automated manner for all catchments. For this reason, we defined the model calibration and evaluation periods as comprising days with significant snow cover on the catchment (>25% of the catchment covered by snow). We included a full week after the occurrence of snowmelt to account for runoff delay. We obtained the value of 25% through empirical tests on the number of days with specific snow coverage values and their corresponding snow water equivalent values for each catchment. We found that below this value the total snow water equivalent in the studied catchments usually becomes negligible.

We calibrated the model for all the catchments in the study using a split-sample approach. We selected this approach because it allowed us to assess both the best possible model performance with respect to each objective function for each model structure variant (i.e. calibration period), and a realistic model application scenario (i.e. validation period), helping us to distinguish between real model improvement and overfitting. In our case, the simulation period was limited by the input data with the shortest temporal availability, which in this case was the snow water equivalent data for the Swiss catchments. In total

20 years were available, which we divided into two equally long 9-year periods plus 2 years for model warm-up. We calibrated the model for both periods and cross-validated the simulations on the remaining periods. For the Swiss catchments, we used the period between 1st of September 1998 and 31st August 2016, while for the Czech catchments we used the period between 1st November 1996 and 31st October 2014. The different start dates for simulation periods in the Swiss and Czech catchments correspond to the different timing for the onset of snow conditions in the different areas. Additionally, the different years included in each study domain correspond to data limitations in each area. Since the two areas were quite distant, we considered that it was more important to have the same period length for running the simulations in both domains rather than using the exact same years, as the meteorological conditions are different in the two study domains anyway.

We calibrated the model for all possible combinations of the single modifications to individual components of the snow routine of the HBV model described in Section 2.2 (n = 64), catchments (n = 54), simulation periods (n = 2), and objective functions (n = 2) using a genetic algorithm (Seibert, 2000). Every calibration effort consisted of 3500 model runs with constrained parameter ranges based on previous studies (Seibert, 1999; Vis et al., 2015). We performed ten independent calibrations for each setup to be able to capture the uncertainty of the model. In total we performed approximately 500 million model simulations. To to assess the impact of potential equifinality and parameter uncertainty issues, we performed a Monte Carlo sensitivity analysis on the all calibration parameters for each for the model structure variants.

## 3 Results

The large number of catchments and model variations considered in this study made it challenging to grasp any details when looking at the entire dataset. For this reason, we first present the results for a single catchment to explore the implications of individual model modifications and illustrate the general trends observed across the study domain. For this purpose we selected the Allenbach catchment at Adelboden (CH-204), one of the high altitude, snow-dominated catchments in the set, as sample catchment. Thereafter, we progressively add more elements to the analysis of the results. Additionally, even if we calibrated (and validated) the model for both periods defined in the split-sample test, here we only present the results for the calibration effort in period 1 and corresponding model validation in period 2, as they are representative for the entire analysis. A comprehensive list including calibration and validation model performance values for both objective functions and all catchments included in this study can be found in Appendix A.

The calibration performance of the standard HBV model for the Allenbach catchment was satisfactory for both objective functions (model performance values of ~0.90) but, still, some modifications led to increased model performances. Amongst the different changes in single components of the snow routine structure of the HBV model that we evaluated in this study, using a seasonally varying degree-day factor ($C_{0,s}$) had the most substantial impact on the model performance to represent snow water equivalent followed by, to a lesser extent, stream runoff (Figure 3). Apart from this modification, only the use of an exponential function to define the precipitation partition between rain and snow ($\Delta P_e$) produced significant changes in the model performance against both objective functions. In this case, however, this modification impacted the model performance

in opposite ways, leading to decreased model performance for the calibration against stream runoff. Model uncertainty, as given by the performance ranges obtained when aggregating the different calibration efforts, was small when compared to the

performance differences between the different model structures.

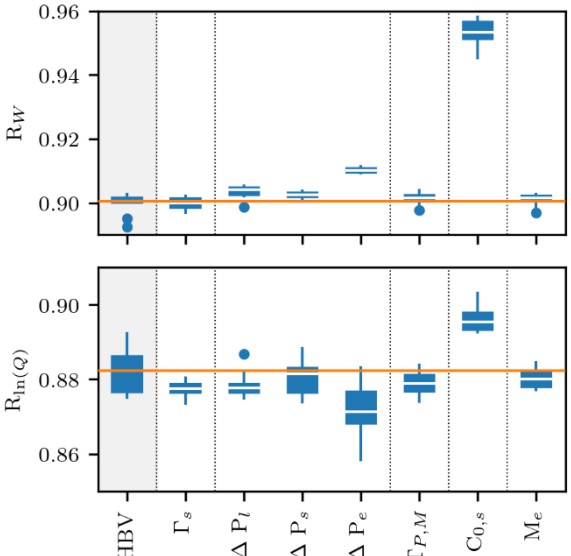

**Figure 3 Model calibration performance for the 10 calibration efforts against the two objective functions (top: snow water equivalent; bottom: logarithmic stream runoff) for each of the modifications to individual components of the snow routine of the HBV model for the Allenbach catchment at Adelboden (CH-204). The modifications include a seasonally-variable temperature lapse**

**rate ($\Gamma_s$), a linear, sinusoidal, and exponential function for the precipitation phase partition ($\Delta P_l$, $\Delta P_s$, and $\Delta P_e$ respectively), different thresholds for precipitation phase and snowmelt ($T_{P,M}$), a seasonally variable degree-day factor ($C_{0,s}$), and an exponential snowmelt with no refreezing ($M_e$). The median performance of HBV is represented with an orange horizontal line.**

Looking at a sample year within the calibration period, we can get a grasp on how comparable the simulated values of snow water equivalent and stream runoff (including the model uncertainty) are to the observed values (Figure 4). While capturing

the general evolution of the snowpack, the HBV model tended to underestimate the snow water equivalent amounts, except for the spring snowmelt period. The model alternative using a seasonal degree-day factor ($C_{0,s}$), which had proven to be the best possible model structure modification for model calibration against snow water equivalent for this catchment, exhibited the same overall behaviour but being more accurate and precise than the HBV model. Regarding the calibration against stream runoff, both model alternatives performed well for low flow periods, but they missed or underestimated some of the peaks.

Model uncertainty was comparable for both model alternatives and was not significant when compared to the simulated values. Model results from the same sample year for all the catchments included in this study can be found in the Supplement.

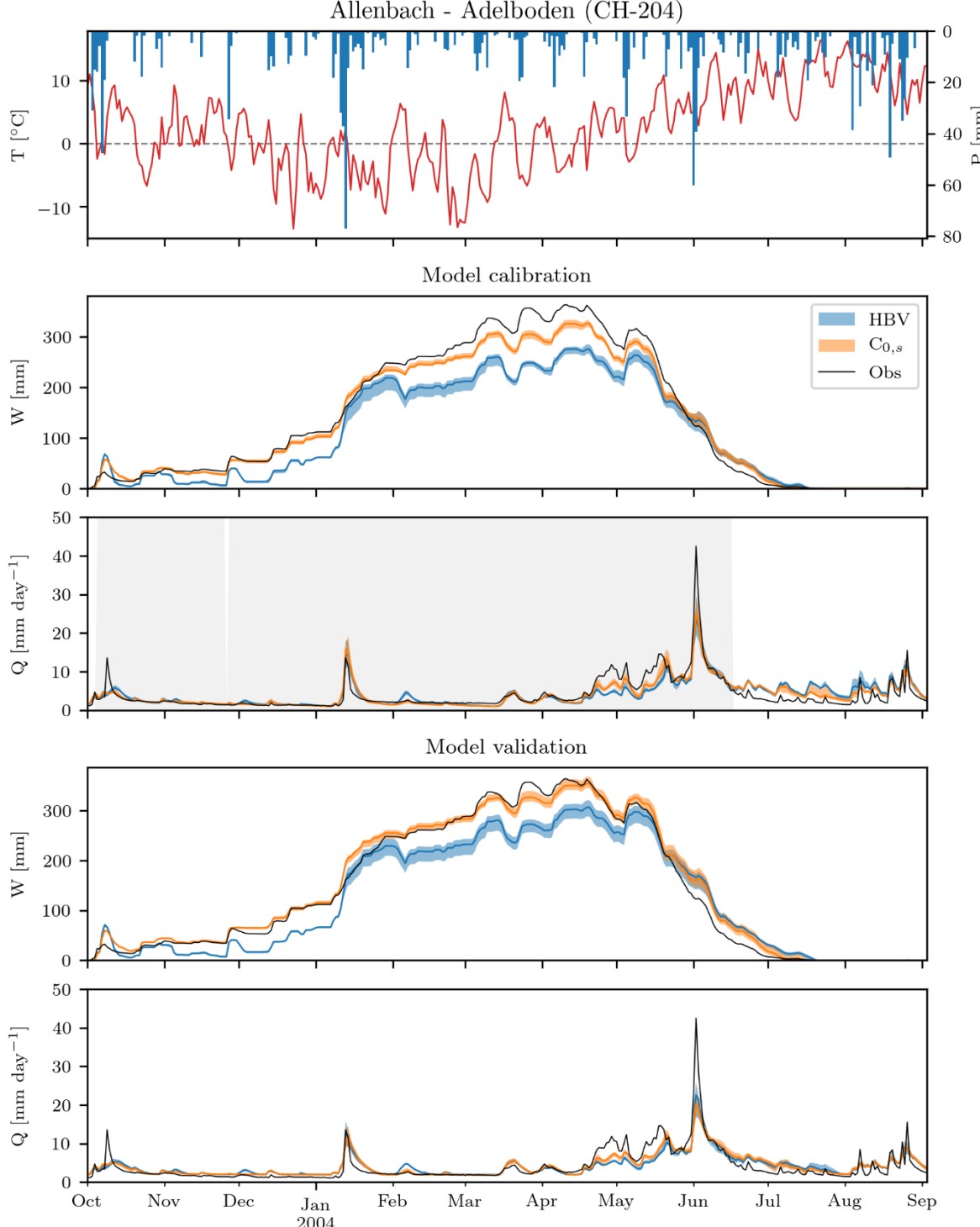

**Figure 4 Example time series (September 2003 – August 2004) from the Allenbach catchment at Adelboden (CH-204). Top: daily mean air temperature and total precipitation. Middle: model calibration results. Bottom: model validation results. The model calibration and validation are further subdivided into (top) catchment-average observed (grey line) and simulated snow water equivalent (HBV in blue and the model structure modification including a seasonally-varying degree-day factor, $C_{0,s}$ in orange), and (bottom) observed (grey line) and simulated stream runoff (HBV in blue and the model structure modification including a seasonal degree-day factor in orange). The grey field represents the period used when calibrating the model against the logarithmic stream runoff. The uncertainty fields for model simulation cover the $10^{th} – 90^{th}$ percentiles range while the solid line represents the median value.**

Looking at the entire set of catchments, the impact of the different model structure modifications on model calibration performance was generally more pronounced for $R_W$ than for $R_{ln(Q)}$ across all catchments (Figure 5). For most catchments, the largest model performance improvements when calibrating against snow water equivalent were achieved by using a seasonally variable degree-day factor ($C_{0,s}$). Using different thresholds for precipitation phase partition and snowmelt ($T_{P,M}$) and using an exponential function for precipitation phase partition ($\Delta P_e$) also conveyed a significant improvement for some of the catchments. Nevertheless, the latter modification performed almost equal to the HBV model when calibrating against stream runoff, and even slightly worse for some catchments. Using an exponential function to define the precipitation partition between rain and snow consistently penalised the model performance when calibrating the HBV model against stream runoff, whereas using an exponential function for snowmelt ($M_e$) was the best alternative when calibrating the model against this objective function. Overall, most modifications conveyed slight model performance improvements concerning snow water equivalent simulations for most of the catchments in the dataset. Nevertheless, the modifications on the precipitation phase partition tended to penalise most Czech catchments when calibrating against snow water equivalent. We did not observe any significant connection between model performance and catchment characteristics such as mean catchment elevation, catchment area, or yearly snowmelt contribution to runoff.

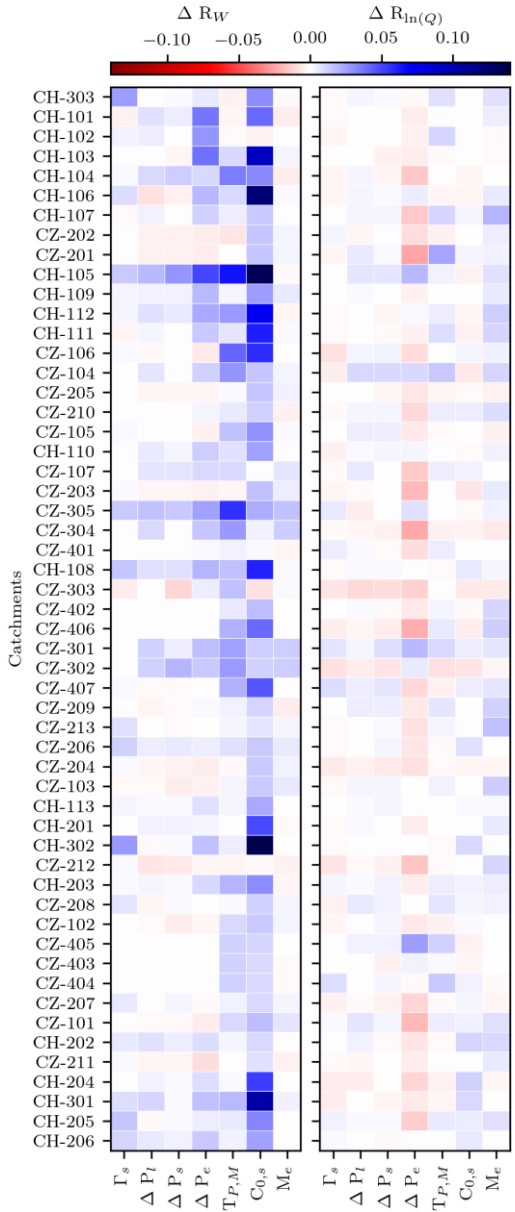

**Figure 5 Median relative model calibration performance for alternative HBV model structures, including modifications to single components of the snow routine with respect to HBV. The modifications include a seasonally-variable temperature lapse rate ($\Gamma_s$), a linear, sinusoidal, and exponential function for the precipitation phase partition ($\Delta P_l$, $\Delta P_s$, and $\Delta P_e$ respectively), different thresholds for precipitation phase and snowmelt ($T_{P,M}$), a seasonally variable degree-day factor ($C_{0,s}$), and an exponential snowmelt with no refreezing ($M_e$). Left: model calibration against snow water equivalent; right: model calibration against logarithmic stream runoff. The catchments are ordered by mean yearly snowmelt contribution to runoff in downwards increasing order.**

While some modifications had a clear and consistent impact on model calibration performance in all catchments, most of them presented a less pronounced either positive or negative impact, depending on the catchment, making it difficult to evaluate

which model structures were more suitable than others (including the default HBV structure) for most of the catchments. Additionally, to better understand the usefulness of the different modifications in real applications, we need to take into account which of model structures performed best for the validation period as well (Figure 6). As already observed in Figure 5, using a seasonal degree-day factor ($C_{0,s}$) was the best modification for calibrating the model against snow water equivalent for the vast majority of the catchments. Nevertheless, this modification ranked relatively low when validating the model against the same objective function. Looking at stream runoff, using an exponential function for snowmelt simulation while disregarding the refreezing process ($M_e$) was the best-ranking modification for both model calibration and validation while the HBV model ranked higher than several of the considered modifications. Using an exponential function to define the precipitation partition between rain and snow ($\Delta P_e$) was the worst alternative for calibrating the model against stream runoff. The diagonal pattern from the top left to the bottom right observed for model calibration indicates that modifications tended to have the same rank for most catchments (note that the ranking of modifications is different when looking at snow water equivalent with respect to stream runoff). Such a pattern was not present for model validation, suggesting that, in that case, there were no model structures significantly more suitable than others.

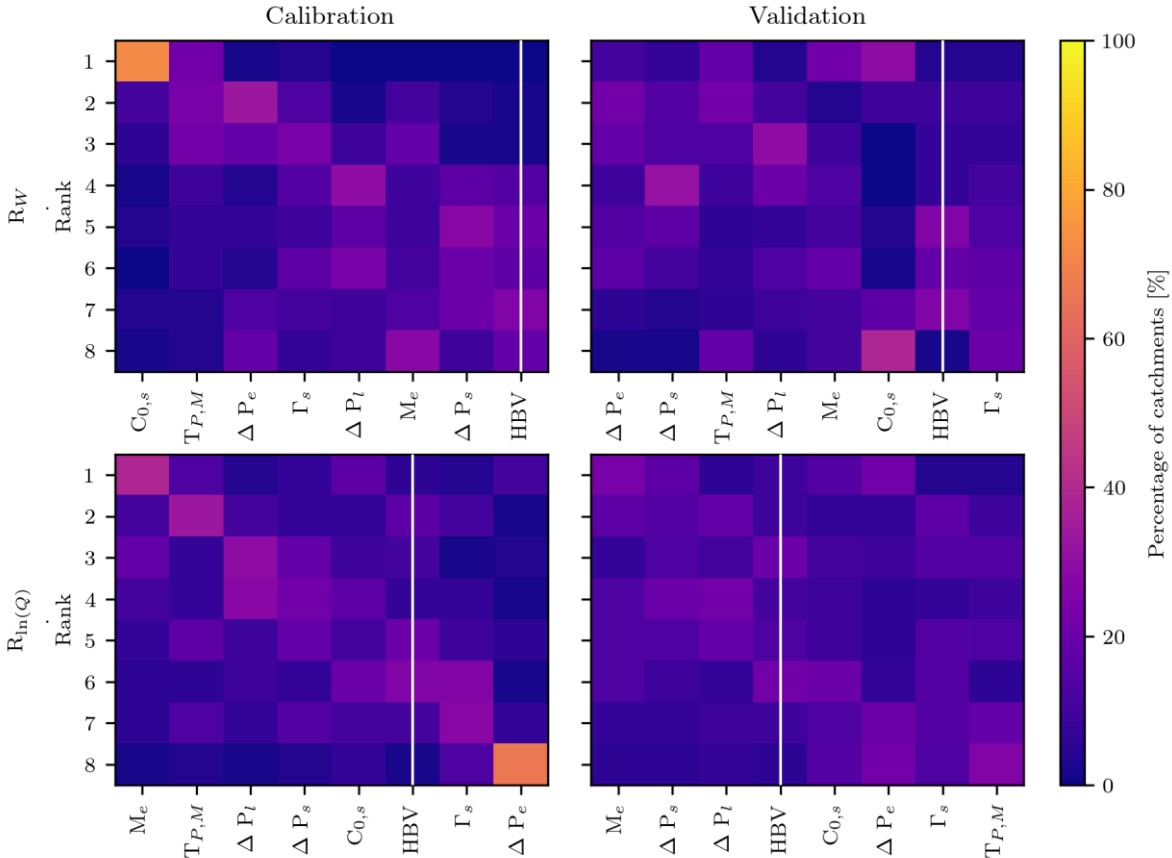

**Figure 6 Rank matrices for each of the model simulation scenarios. Top: model calibration (left) and validation (right) against snow water equivalent; bottom: model calibration (left) and validation (right) against logarithmic stream runoff. Each rank matrix shows the rank distribution of each modification to single components of the snow structure of the HBV model for all the catchments included in this study so that each column adds to 100%. The modifications are ordered from highest to lowest average ranking (left to right) and include a seasonally-variable temperature lapse rate ($\Gamma_s$), a linear, sinusoidal, and exponential function for the precipitation phase partition ($\Delta P_l$, $\Delta P_s$, and $\Delta P_e$ respectively), different thresholds for precipitation phase and snowmelt ($T_{P,M}$), a seasonally variable degree-day factor ($C_{0,s}$), and an exponential snowmelt with no refreezing ($M_e$). The HBV model structure is highlighted with a white vertical line.**

Even if some alternative model structures clearly improved the calibration performance of the model, most structures had a limited impact on model performance. This is in part because, to this point, we only tested model structures containing a single modification with respect to the HBV model. We next explored whether the same trends persisted when including further elements to the model by using an increasing amount of model structure modifications simultaneously. In, total we tested 64 different model structures (Table 3). Using the maximum possible number of simultaneous modifications (5) to the model structure would result in the use of up to nine additional parameters. This would lead to a clear overparameterisation of the model (the default snow routine structure of HBV contains five parameters), but we included this alternative to provide a complete analysis of the all available alternatives.

**Table 3 Number of model structure alternatives containing a given number of snow routine modifications.**

| Number of modifications | Number of alternatives |
|:---:|:---:|
| 0 | 1 |
| 1 | 7 |
| 2 | 18 |
| 3 | 22 |
| 4 | 13 |
| 5 | 3 |
| | n = 64 |

Figure 7 shows the median model performance for each of the 64 possible model structure alternatives for all catchments relative to the standard HBV model performance, sorted by the number of components being modified. When calibrating the model against snow water equivalent, model performance clearly increased for all of the model structure alternatives. The impact was more modest for model validation with a significant percentage of alternative structures performing worse than HBV. Regarding model calibration against stream runoff, the effect of an increasing number of components being modified was limited but mostly positive. The range of model performance values was also significantly smaller than when looking at snow water equivalent. This relates to the fact that, for most catchments, the snow routine has a limited weight over the entire HBV model. For model validation we observed a similar trend, but with broader model performance ranges. Also, the fact that performance variability varied significantly with the number of components being modified was in part due to the differences

in the number of model structure alternatives for each of them, being larger for those number of modifications which included

the largest amount of model structure alternatives.

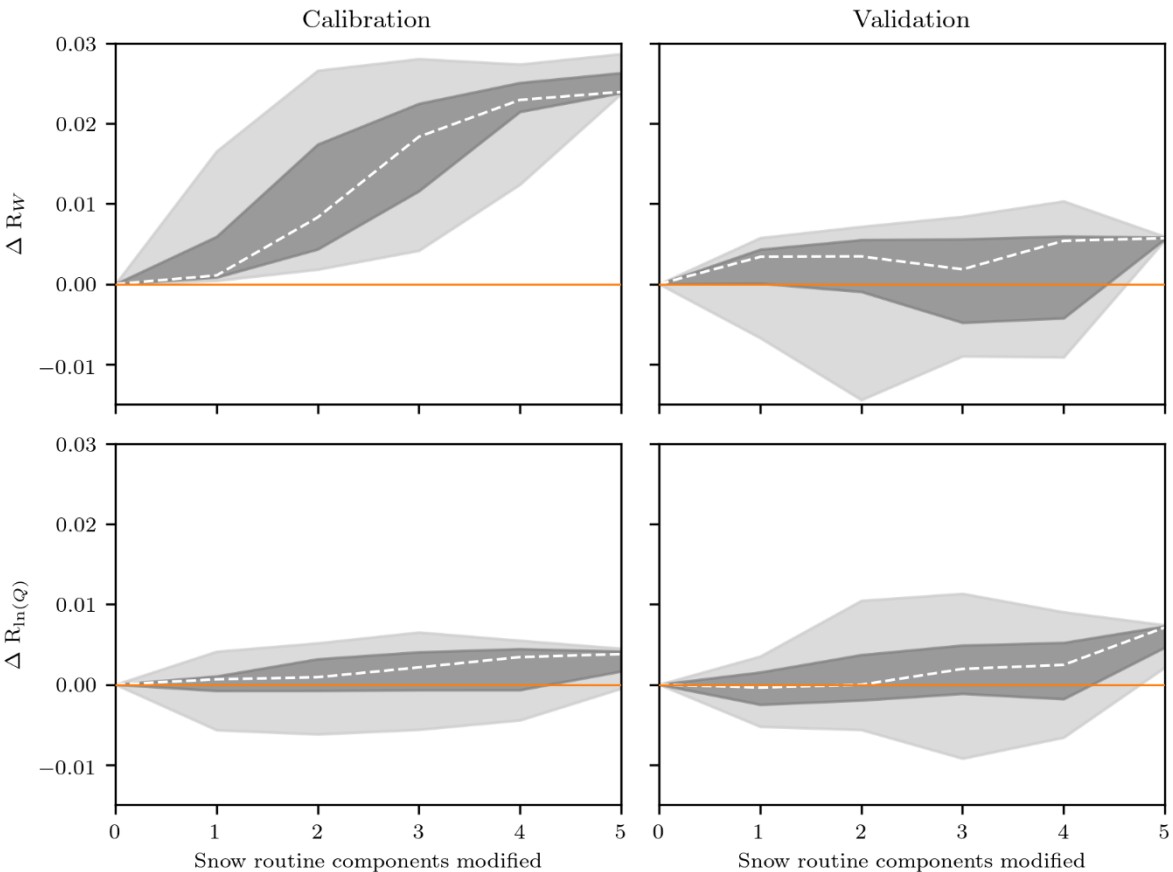

**Figure 7 Median relative model performance with respect to HBV across all catchments for the 64 considered snow routine structures sorted by an increasing number of modifications. Model simulations against snow water equivalent are presented in the 450   top row, while those against logarithmic stream runoff are shown in the bottom row. Model calibration is presented in the left column and model validation in the right one. The dashed line represents the median value across all catchments while the grey fields represent the minimum to maximum (light grey) and 25th to 75th percentiles (dark grey). The relative model performance of HBV is highlighted with a solid orange line.**

In general, we observed an increase in model performance for all cases. However, except for model calibration against snow

water equivalent, there was no clear indication that a model with a more detailed formulation of the snow processes would

lead to significantly improved model performance. Indeed, the range of performances among the different model structures

was larger than the median net increase. This might be an indication that choosing the right modifications (or combinations of

modifications) is more relevant than significantly increasing model detail. This way, we attempted to determine whether some

specific modifications conveyed a model performance gain across all model structures in which they were included for both

model calibration and validation against the two objective functions. To this purpose, we ranked all model structures for model

calibration and validation against both objective functions and visualised the cumulative distribution of each of the individual

model modifications (Figure 8). Some of the patterns observed here resemble those that we already observed for single modifications only (Figure 6). For instance, all top-ranking model structures included a seasonally variable degree-day factor ($C_{0,s}$) for model calibration against snow water equivalent. Similarly, all bottom-ranking model structures used an exponential

function for precipitation phase partition ($\Delta P_e$) for model calibration against stream runoff. Besides these familiar patterns, other patterns emerged, which could not be clearly observed when only looking at single modifications. Indeed, even if a seasonal degree-day factor performed above average in most cases, this particular modification was included in all of the bottom-ranking model structures for model validation against snow water equivalent. Additionally, model structures including an exponential function for snowmelt ($M_e$) performed above average for all cases, and were even included in almost all the

top-ranking model structures.

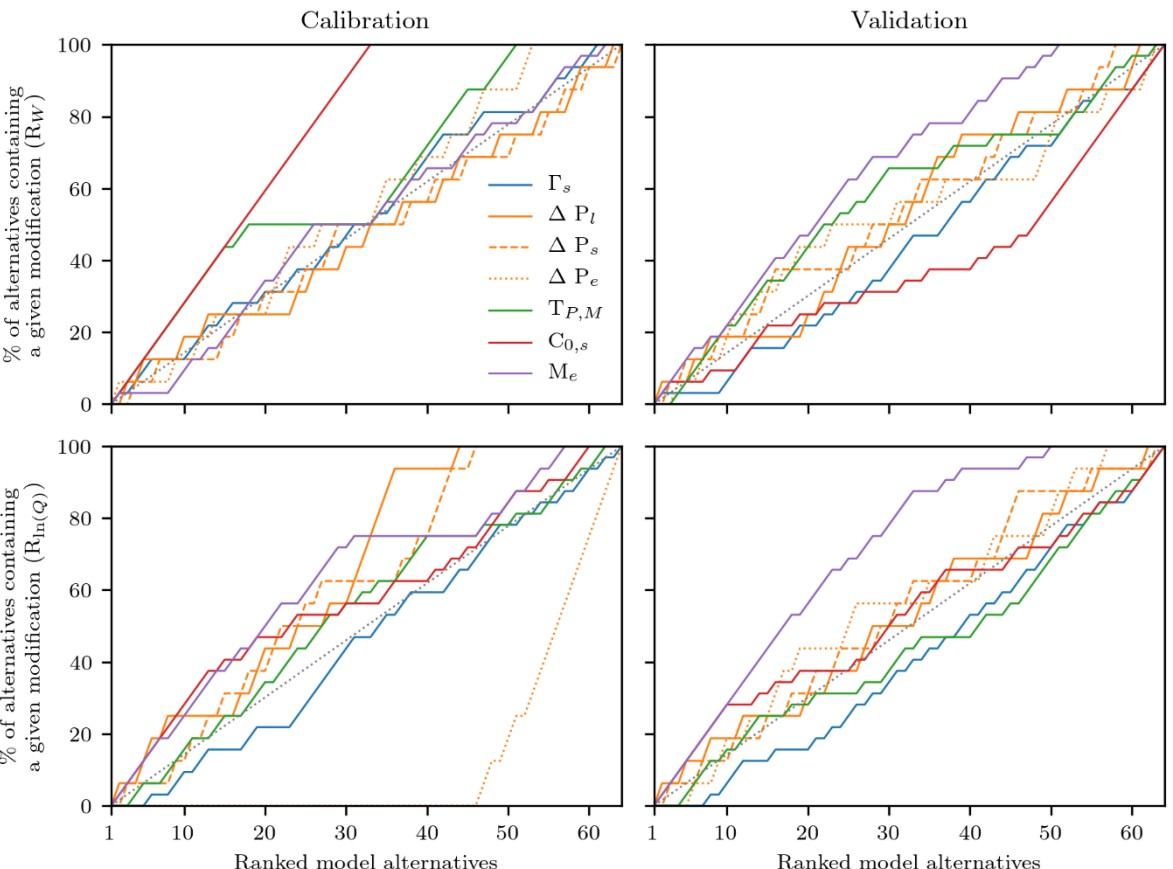

**Figure 8 Cumulative plots for each of the 7 individual modifications to the snow routine of the HBV model as a function of the ranked 64 model structures arising from all the possible combinations of modifications. The modifications are a seasonally-variable temperature lapse rate ($\Gamma_s$), a linear, sinusoidal, and exponential function for the precipitation phase partition ($\Delta P_l$, $\Delta P_s$, and $\Delta P_e$**

**respectively), different thresholds for precipitation phase and snowmelt ($T_{P,M}$), a seasonally variable degree-day factor ($C_{0,s}$), and an exponential snowmelt with no refreezing ($M_e$). Model simulations against snow water equivalent are presented in the top row, while those against logarithmic stream runoff are presented in the bottom row. Model calibration is presented in the left column and model validation in the right one. Model modifications plotting above the 1:1 line (grey dotted line) tend to be included in high-ranking model structures, while those plotting below the 1:1 line tend to be included in low ranking structures.**

Based on the ranked alternative model structures and the modifications contained in each of them, specific model modifications contributed the most to model performance increases, regardless of other model structure modifications. Nevertheless, these dominant modifications impacted the model structure performance in different ways, depending on the modelling scenario (i.e. calibration/validation, objective function). Ideally, any model structure modifications should convey an improved representation of snow water equivalent but also have a positive impact on the simulation of stream runoff (which is the main

output of the model), both for model calibration and validation.

To achieve an improved representation of both snow water equivalent and stream runoff we only took into account those model structures that led to a positive impact for each of the four modelling scenarios (i.e. calibration and validation efforts against both objective functions). This way, we selected and ranked all model alternatives that had a positive median relative model performance value with respect to the HBV model and examined which modifications led to the largest model performance

improvement (Figure 9). All of the selected model structures contained an exponential function for snowmelt ($M_e$), and none of them included an exponential function for precipitation phase partition ($\Delta P_e$). Most model structures were the result of the combination of three to four individual model structure modifications (seven model structures each). Four model structures contained two model modifications and two model structures contained five modifications. Perhaps most interestingly, two of the model structures included only a single modification: an exponential snowmelt function, and a sine function for

precipitation phase partition ($\Delta P_s$). Nevertheless, these alternatives had the lowest ranking amongst the selection. Overall, the top-ranking alternatives contained a seasonally varying degree-day factor ($C_{0,s}$) and an exponential snowmelt function, while other individual modifications resulted in more considerable model performance variability.

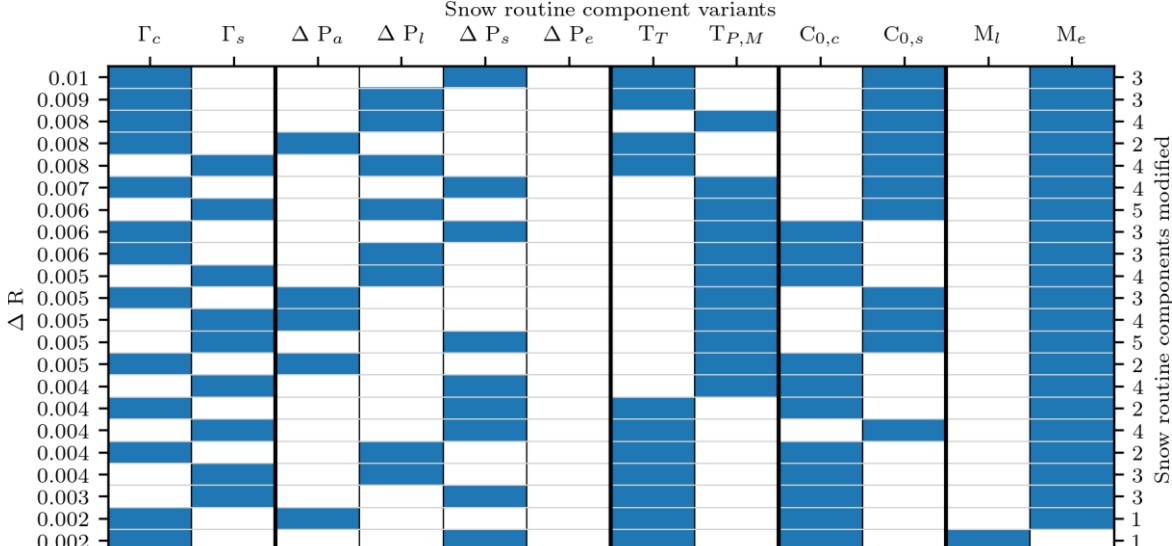

**Figure 9 Ranked alternative structures to the snow routine of HBV that present positive relative model performance values with**
500 **respect to HBV for model calibration and validation against snow water equivalent and stream runoff disaggregated by snow routine component variants. The alternatives for each of the considered model components are a linear and a seasonally-variable degree-day factor ($\Gamma_c$ and $\Gamma_s$ respectively); an abrupt, linear, sinusoidal, and exponential precipitation phase partition ($\Delta P_a$, $\Delta P_l$, $\Delta P_s$, and**

*ΔP$_e$ respectively); a common and individualised threshold temperature for precipitation phase partition and snowmelt ($T_T$ and $T_{P,M}$ respectively); a constant and seasonally-variable degree-day factor ($C_{0,c}$ and $C_{0,s}$ respectively); and a linear and exponential (with no refreezing) melt function ($M_l$ and $M_e$ respectively). Every row contains one model structure with the selected variant for each of the components highlighted in blue. The median relative model performance for all modelling scenarios is given on the left y-axis while the number of model modifications in each alternative is provided in the right y-axis.*

While not shown explicitly in this paper, the results obtained from the Monte Carlo sensitivity analysis showed that, even if some of the model structure variants (i.e. $T_{p,m}$, $\Delta P_e$, or $C_{0,s}$) produced compensating effects on some of the model parameters (e.g. precipitation lapse rate, maximum storage in soil box, threshold for reduction of evaporation, or the shape coefficient), this effect was only observed for a reduced number of catchments. Most parameters showed no compensating effects at all. Overall, parameter values and their sensitivity tended to be reasonably consistent across all the tested model structure variants for most of the catchments in the study.

## 4 Discussion

It is challenging to improve existing rainfall-runoff models and especially those that, like HBV, have successively been tested and applied in many catchments over a range of environmental and geographical conditions (Bergström, 2006). Nevertheless, some of the proposed alternative snow routine model structures that we tested in this study showed a generally positive impact on model performance for simulating both snow water equivalent and stream runoff, albeit to different extents. We found that the most valuable modification to single HBV snow routine components for rainfall-runoff modelling in mountainous catchments in Central Europe was the use of an exponential snowmelt function and, to a lesser extent, a seasonally-varying degree-day factor. Another modification, using different thresholds for snowfall and snowmelt instead of a single threshold, produced a significant model performance improvement regarding snow water equivalent, but did not convey any advantage for simulating stream runoff.

We observed a significant differences in model performance changes between both objective functions when testing the different snow routine model structures. Indeed, in general, the impact was more evident when simulating snow water equivalent than when simulating stream runoff, as the latter is the result of the combined model routines(i.e. snow, soil, groundwater, and routing routines), which partially compensate and mask any modifications made on the snow routine (Clark and Vrugt, 2006). Additionally, some of the modifications that improved the model performance against snow water equivalent, such as the use of an exponential function to define the solid and liquid phases of precipitation, resulted in poorer the stream runoff simulations.

Unlike most modifications considered in this study, which are simple conceptualisations of complex processes, the use of an exponential function to describe precipitation phase partition and the use of a seasonally varying temperature lapse rate are both formulations derived from empirical evidence (Magnusson et al., 2014; Rolland, 2002). Nevertheless, as we have previously discussed, neither of these modifications translate into an improvement of model performance for either objective functions. This might be because, since conceptual models such as HBV are based on simplifications and generalisations of

the processes that occur in reality, formulations based on accurate measurements of diverse processes do not align well with the other simplifications made in the model structure and/or behaviour at the chosen spatio-temporal resolution (Harder and Pomeroy, 2014; Magnusson et al., 2015).

Other modifications are relatively similar to each other, such as the case of using linear and sine functions to describe precipitation phase partition. Both these formulations require only one additional parameter, and perform almost identical: the precipitation partition between rain and snow is exactly the same for both formulations for most of the transition temperature range except for the tails, which are abrupt for the linear case and smooth for the sine one. Provided that the smooth transition is a more accurate description of the physical process which, in addition, avoids the introduction of discontinuities into the objective functions – which might complicate model calibration (Kavetski and Kuczera, 2007) –, and that both modifications include the same number of parameters and perform nearly identical, the most accurate description should be preferred. Nevertheless, some models, including HBV, continue to use the linear conceptualisation with the argument of simplicity.

Even if we did not observe differences in model performance as a function of different catchment characteristics, snow water equivalent tends to be underestimated in lowland catchments, while we observed no clear pattern for alpine catchments. Nevertheless, the limited number of high-elevation catchments in the dataset (only four of them have a mean elevation above 2000 m a.s.l.) combined with the generally small size, steep topography, relatively large glacierised areas, scarce vegetation and exposure to extreme weather conditions (such as strong wind gusts) of these catchments, makes it difficult to extract any relevant trends. That being said, in general, the different model structures tended to underestimate snow accumulation and delay the timing of the spring snowmelt season. Similar patterns have also been observed for other mountainous areas of Central Europe (Sleziak et al., 2020). We observed differences in model performance among the two geographical domains included in this study.

Among the different snow routine model structures, the modifications on precipitation phase partition penalised the model performance on most Czech catchments for simulating snow water equivalent, while having the opposite effect for Swiss catchments. The Czech catchments have a narrower elevation range compared to the Swiss catchments, in addition to an earlier and shorter snowmelt period. These characteristics may favour the simplification of an abrupt transition between rain and snow, while using gradual transitions between rain and snow might favour the more extended melt season and larger elevation ranges of the Swiss catchments. Another factor that may impact the results is the significant differences in model driving and validation data availability for each of the geographical domains (Günther et al., 2019; Meeks et al., 2017). Indeed, while in Czechia there were a limited number of meteorological stations providing temperature, precipitation and snow water equivalent data, the Swiss catchments benefited from distributed data for the different catchment elevations, allowing for more accurate calibration of snow-related parameters. The difference in resolution between the Swiss and Czech input data might affect the obtained results, where each has its strengths and weaknesses. For instance, high-resolution data can become highly uncertain for individual grid cells while observational data may be affected by measurement errors and representativeness issues. Even so, the model performance variability of the different snow routine structures relative to the default HBV model should be similar for both cases. Overall, even with the large differences in hydrological regime, catchment morphology, and data

availability between the two geographical domains, the impact of the different snow routine model structures on model performance was in general comparable among them.

Regarding model complexity and uncertainty, we found that increasing the degree of detail, and thus the number of parameters of the model, generally translated into a broader range of model performance values, indicating that the uncertainty related to the model structure increased as well. This is a well-known problem of conceptual rainfall-runoff models, and the focus of many studies (Essery et al., 2013; Strasser et al., 2002). Additionally, we found that, for most cases, the median model performance increase with an increasing degree of detail was not significant with respect to the performance range. This means that a more detailed model does not necessarily translate into better model performance, which is consistent with previous studies (Orth et al., 2015). This fact highlights the importance of carefully choosing the degree of detail of the model based on the desired objectives and available data (Hock, 2003; Magnusson et al., 2015). Another important aspect is the uncertainty and robustness of the model's parameters. In conceptual models, such as the HBV model, model parameters can compensate for each other, which makes the interpretation of any model structure modifications rather challenging (Clark and Vrugt, 2006). Nevertheless, a Monte Carlo sensitivity analysis on the HBV model parameters showed that parameter values and sensitivity were consistent for most catchments and model structures in this study.

Even if an increased degree of detail in the processes description is not desirable by itself, as it can lead to overparameterisation and equifinality issues, it can also improve the performance of rainfall-runoff models if it responds to specific needs and/or available data, among others. Indeed, 22 of the 63 model structure alternatives that we tested in this study (all of them conveying an increase of model detail, and therefore an increase in the number of parameters) convey and increase of model performance with respect to HBV for both model calibration and simulation against both objective functions. Furthermore, out of these 22 alternatives, only two of them consist of a single model structure modification, while most have 3 or 4 modifications. Nevertheless, all of these alternatives share common traits, such as using an exponential snowmelt function with no refreezing. Almost all model structures that do not have this particular modification perform worse than HBV in at least one simulation scenario.

It is reasonable to state that, while the increased degree of detail arising from the interplay among the different model structure modifications play a role in improving the model performance, this is mainly the result of a few dominant modifications. This way, the use of an exponential snowmelt function is the most valuable single modification with a median performance increase of 0.002 for all simulations (with individual performance increases over 0.1). However, when we combine it with a seasonal degree-day factor, we achieve a median performance increase of 0.008, almost the highest performance increase amongst all model alternatives. Adding further detail to the model does not convey significant improvements for this model structure. Consequently, if we were to implement any modifications to the model, they would be to substitute the linear snowmelt and refreezing conceptualisation by an exponential snowmelt function, and replacing the constant degree-day factor by a seasonally varying one, in that order.

Finally, it is important to mention that these results are only valid for the selected study areas and cannot be extrapolated to all the different alpine and snow-covered regions around the world as the different processes involved in different geologic, geographic, climatological, and hydrological settings are likely to favour different formulations of the snow processes.

## 5 Conclusions

We evaluated the suitability of different temperature-based snow routine model structures for rainfall-runoff modelling in Alpine areas of Central Europe. More specifically we tested a number of modifications to each of the components of the snow routine of the HBV model over a large number of catchments covering a range of geographical settings and different data availability conditions based on their ability to reproduce both snow water equivalent and stream runoff. We found that the results differ greatly across the different catchments, objective functions, and simulation type (i.e. calibration/validation). Still they allow drawing the following general conclusions regarding the value of the different snow routine model structures:

- The comparatively simple default structure of the HBV model performs well for simulating snow-related processes and their impact to stream runoff in most of the examined catchments.
- Specific modifications to the formulation of certain processes in the snow routine structure of the model improve the performance of the model for estimating snow processes and, to a lesser extent, for simulating stream runoff.
- An exponential snowmelt function with no refreezing is the single most valuable overall modification to the snow routine structure of HBV, followed by a seasonally variable degree-day factor.
- Adding further detail to the snow routine model structure does not, by itself, add any value on the ability of the model to reproduce snow water equivalent nor stream runoff. A careful examination of the design choices of the model for the given application, data availability, and purpose – such as the one presented in this study – is crucial to ensure that the model conceptualisation is suitable and to provide guidance on potential model improvements.

The specific results obtained in this study are not transferrable to other geographical domains, models, or purposes. Nevertheless, the methodology presented here is relevant to the general degree-day approach. It may, therefore, be used to assess the suitability of model design choices in temperature-based snow routines in other rainfall-runoff models in different circumstances.

## Appendix A: HBV model performance

**Table A1 HBV median calibration and validation performance values – both for snow water equivalent, $R_W$, and logarithmic stream runoff, $R_{lnQ}$ – for each catchment and analysis period.**

| | Period 1 | | | | Period 2 | | | |
|---|---|---|---|---|---|---|---|---|
| ID | $R_W$ | | $R_{lnQ}$ | | $R_W$ | | $R_{lnQ}$ | |
| | Calibration | Validation | Calibration | Validation | Calibration | Validation | Calibration | Validation |

| | | | | | | | |
|---|---|---|---|---|---|---|---|
| CH-101 | 0.75 | 0.68 | 0.92 | 0.87 | 0.82 | 0.73 | 0.87 | 0.81 |
| CH-102 | 0.59 | 0.5 | 0.85 | 0.8 | 0.8 | 0.61 | 0.87 | 0.83 |
| CH-103 | 0.74 | 0.57 | 0.91 | 0.89 | 0.84 | 0.69 | 0.86 | 0.84 |
| CH-104 | 0.77 | 0.69 | 0.83 | 0.76 | 0.81 | 0.79 | 0.75 | 0.63 |
| CH-105 | 0.75 | 0.73 | 0.78 | 0.75 | 0.8 | 0.78 | 0.74 | 0.69 |
| CH-106 | 0.77 | 0.65 | 0.85 | 0.82 | 0.84 | 0.69 | 0.79 | 0.75 |
| CH-107 | 0.81 | 0.78 | 0.85 | 0.82 | 0.81 | 0.78 | 0.83 | 0.8 |
| CH-108 | 0.84 | 0.81 | 0.87 | 0.85 | 0.84 | 0.75 | 0.83 | 0.81 |
| CH-109 | 0.84 | 0.79 | 0.88 | 0.84 | 0.81 | 0.73 | 0.84 | 0.77 |
| CH-110 | 0.92 | 0.9 | 0.87 | 0.85 | 0.91 | 0.86 | 0.84 | 0.82 |
| CH-111 | 0.8 | 0.79 | 0.89 | 0.88 | 0.8 | 0.78 | 0.84 | 0.81 |
| CH-112 | 0.81 | 0.74 | 0.87 | 0.85 | 0.82 | 0.74 | 0.82 | 0.79 |
| CH-113 | 0.92 | 0.91 | 0.86 | 0.85 | 0.88 | 0.86 | 0.85 | 0.84 |
| CH-201 | 0.8 | 0.81 | 0.81 | 0.78 | 0.78 | 0.74 | 0.74 | 0.69 |
| CH-202 | 0.92 | 0.86 | 0.89 | 0.86 | 0.92 | 0.87 | 0.85 | 0.8 |
| CH-203 | 0.88 | 0.87 | 0.9 | 0.86 | 0.84 | 0.83 | 0.9 | 0.87 |
| CH-204 | 0.9 | 0.87 | 0.88 | 0.72 | 0.85 | 0.82 | 0.83 | 0.56 |
| CH-205 | 0.92 | 0.91 | 0.85 | 0.82 | 0.9 | 0.89 | 0.89 | 0.86 |
| CH-206 | 0.9 | 0.87 | 0.94 | 0.92 | 0.85 | 0.82 | 0.95 | 0.92 |
| CH-301 | 0.78 | 0.69 | 0.94 | 0.92 | 0.91 | 0.84 | 0.95 | 0.94 |
| CH-302 | 0.7 | 0.62 | 0.94 | 0.92 | 0.87 | 0.83 | 0.94 | 0.93 |
| CH-303 | 0.77 | 0.71 | 0.92 | 0.7 | 0.83 | 0.64 | 0.91 | 0.86 |
| CZ-101 | 0.86 | 0.77 | 0.84 | 0.81 | 0.92 | 0.79 | 0.82 | 0.65 |
| CZ-102 | 0.86 | 0.76 | 0.88 | 0.82 | 0.92 | 0.78 | 0.85 | 0.74 |
| CZ-103 | 0.86 | 0.77 | 0.85 | 0.8 | 0.92 | 0.79 | 0.83 | 0.73 |
| CZ-104 | 0.91 | 0.88 | 0.81 | 0.74 | 0.75 | 0.72 | 0.83 | 0.72 |
| CZ-105 | 0.9 | 0.8 | 0.81 | 0.76 | 0.94 | 0.87 | 0.8 | 0.73 |
| CZ-106 | 0.85 | 0.67 | 0.84 | 0.8 | 0.93 | 0.88 | 0.81 | 0.72 |
| CZ-107 | 0.89 | 0.84 | 0.8 | 0.74 | 0.92 | 0.87 | 0.87 | 0.77 |
| CZ-201 | 0.92 | 0.82 | 0.81 | 0.76 | 0.91 | 0.85 | 0.7 | 0.64 |
| CZ-202 | 0.92 | 0.83 | 0.69 | 0.47 | 0.91 | 0.85 | 0.66 | 0.44 |
| CZ-203 | 0.92 | 0.82 | 0.84 | 0.59 | 0.91 | 0.85 | 0.84 | 0.68 |
| CZ-204 | 0.92 | 0.83 | 0.74 | 0.71 | 0.91 | 0.85 | 0.89 | 0.87 |
| CZ-205 | 0.92 | 0.83 | 0.88 | 0.84 | 0.91 | 0.86 | 0.81 | 0.77 |
| CZ-206 | 0.86 | 0.63 | 0.83 | 0.79 | 0.89 | 0.78 | 0.81 | 0.76 |
| CZ-207 | 0.87 | 0.62 | 0.79 | 0.58 | 0.89 | 0.78 | 0.81 | 0.65 |
| CZ-208 | 0.87 | 0.62 | 0.66 | 0.55 | 0.89 | 0.79 | 0.83 | 0.73 |

| | | | | | | | |
|---|---|---|---|---|---|---|---|
| CZ-209 | 0.95 | 0.92 | 0.79 | 0.7 | 0.93 | 0.88 | 0.82 | 0.77 |
| CZ-210 | 0.95 | 0.92 | 0.81 | 0.74 | 0.93 | 0.89 | 0.81 | 0.6 |
| CZ-211 | 0.87 | 0.62 | 0.83 | 0.8 | 0.89 | 0.78 | 0.85 | 0.75 |
| CZ-212 | 0.88 | 0.62 | 0.79 | 0.74 | 0.9 | 0.8 | 0.83 | 0.76 |
| CZ-213 | 0.87 | 0.64 | 0.81 | 0.77 | 0.9 | 0.79 | 0.82 | 0.8 |
| CZ-301 | 0.93 | 0.87 | 0.81 | 0.7 | 0.97 | 0.92 | 0.86 | 0.78 |
| CZ-302 | 0.93 | 0.87 | 0.84 | 0.71 | 0.97 | 0.92 | 0.85 | 0.67 |
| CZ-303 | 0.94 | 0.87 | 0.86 | 0.8 | 0.97 | 0.91 | 0.87 | 0.77 |
| CZ-304 | 0.93 | 0.87 | 0.89 | 0.86 | 0.97 | 0.92 | 0.87 | 0.82 |
| CZ-305 | 0.79 | 0.67 | 0.82 | 0.7 | 0.81 | 0.75 | 0.83 | 0.74 |
| CZ-401 | 0.95 | 0.89 | 0.84 | 0.81 | 0.9 | 0.85 | 0.78 | 0.69 |
| CZ-402 | 0.92 | 0.9 | 0.74 | 0.7 | 0.93 | 0.92 | 0.82 | 0.77 |
| CZ-403 | 0.92 | 0.89 | 0.81 | 0.78 | 0.96 | 0.94 | 0.86 | 0.82 |
| CZ-404 | 0.91 | 0.89 | 0.75 | 0.73 | 0.96 | 0.94 | 0.82 | 0.8 |
| CZ-405 | 0.91 | 0.89 | 0.79 | 0.76 | 0.96 | 0.93 | 0.87 | 0.84 |
| CZ-406 | 0.85 | 0.82 | 0.78 | 0.71 | 0.92 | 0.86 | 0.8 | 0.66 |
| CZ-407 | 0.85 | 0.82 | 0.73 | 0.67 | 0.92 | 0.87 | 0.74 | 0.7 |
| Median | 0.86 | 0.78 | 0.83 | 0.77 | 0.89 | 0.82 | 0.83 | 0.76 |

## 6 Data availability

Meteorological and hydrological data to calibrate the HBV model were obtained from the Swiss Federal Office of Meteorology and Climatology, the Swiss Federal Office for the Environment, and the Czech Hydrometeorological Institute. The HBV model outputs are available from the first author upon request.

## 7 Author contribution

JS initiated the study. MGL developed the methodology and performed all analyses. MGL, NG, and MJ prepared the input meteorological and hydrological data used to calibrate the HBV model. MJPV performed all the necessary modifications to the source code of the HBV model. MGL prepared the manuscript with contributions from all co-authors.

## 8 Competing interests

The authors declare that they have no conflict of interest.

## 9 Acknowledgements

This project was partially funded by the Swiss Federal Office for the Environment (FOEN). The contribution of Michal Jenicek was supported by the Czech Science Foundation, project no. GA18-06217Y. We thank reviewers Juraj Parajka, Valentina Premier, María José Polo, and Thomas Skaugen for the valuable comments and suggestions that greatly improved this paper.

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
