# Peer review of "Assessing the degree of detail of temperature-based snow routines for runoff modelling in mountainous areas in Central Europe"

_Hydrology and Earth System Sciences, 2020_

## Referee Comment (RC1) · Juraj Parajka (Referee) · 3 Mar 2020

General comments

The study evaluates snow and runoff performance of 64 snow routine alternatives based on degree-day approach in large sample of catchments (54) located in Swiss and Czech Republic. The snow routine variants are coupled with HBV conceptual hydrologic model and model simulations are evaluated in terms of observed daily runoff and snow water equivalent observations/estimates. The results indicate that exponential snowmelt function with no refreezing and seasonally variable degree-day factors are the most reliable/robust/accurate variants for snowmelt runoff simulations in se-

lected catchments.

Overall, this is an interesting study which is worth to publish. The topic is relevant and within the scope of the journal. The study is clearly written and has a good structure. The analyses and interpretations are based on larger sample of catchments which allows to draw interpretations/conclusions that are relevant for large region of similar physiographic conditions in Central Europe.

I have only few comments/notes which can be considered (in my opinion) to add/extend/improve clarity and generality of findings. These include:

1) Perhaps it will be possible to refer here in general to variants of degree-day snow approach, not strictly limits the analysis to HBV variants. The results can be used/implemented in degree-day routines of different hydrological models. In this study, the variants are coupled with HBV concept of rainfall-runoff transformation, but I believe, at least the evaluation of snow efficiency is relevant to general degree-day approach.

2) When coupling the 64 snow routine variants with HBV model, there is another interesting question, which can be discussed and this is the robustness/uncertainty of other HBV model parameters. How consistent/different are the other HBV model parameters for different snow variants? Are, for example, field capacity or nonlinear runoff generation (beta) parameter values similar or compensating some effects of different snow routines?

3) It is not clear which part of the snow accumulation/melt phases are described/evaluated by selected snow objective function? For some practical applications, for example, it will be interesting to see the difference in maximum snow water equivalent between the routines, or to what extent the model over or underestimates snow cover duration? To what extent are these aspects covered in current snow efficiency evaluation? Does a good simulation mean well represented maximum SWE or snow cover duration? Perhaps there are some differences in such efficiency between

the variants.

4) In our recent study (Sleziak et al., 2020) we found that there are quite significant differences in snow model performance (by using standard HBV degree-day approach) between lowland and alpine catchments in Austria. (Differences in terms of overestimation of snow cover in alpine and underestimation of snow cover in flatland catchments). Did you observe similar findings here?

Specific comments

1) Abstract. Is the last sentence needed?

2) Introduction: It will be interesting to extend somewhat this section by refereeing to ways how can be/are degree-day routine parameters estimated in hydrological models.

3) Data: How close are gridded snow water equivalent data to observations? Is there some bias related to the fact that this dataset is based on some type of degree-day model?

4) Runoff model efficiency. Why only Nash-Sutcliffe based on logarithmic transformed discharges? It will be interesting to see also the model performance in terms of snowmelt runoff peaks.

5) P.15, l.355: Figure 3 or Figure 4?

6) Figure 4. Will it be possible to show such case for a year in the validation period?

7) Results: Will it be possible to present runoff and snow model efficiencies for each catchment in the Supplement?

References:

Sleziak et al. (2020) The effect of the snow weighting on the temporal stability of hydrologic model efficiency and parameters, Journal of Hydrology.

[Figure]

57, 2020.
Interactive
comment

---

## Referee Comment (RC2) · Giacomo Bertoldi (Referee) · 3 Mar 2020

In attach some specific comments of Valentina Premier Ph.D. working with me.

The paper applies some modifications on the snow routine of the HBV model. Main results are that an increasing complexity does not lead to increasing performance. The most positively influencing modification is the use of an exponential snowmelt function and of a seasonally variable degree-day factor. Some comments follow: - Line 15-17: "However, [. . .] support tool" This sentence is not really clear to me. In general, I would restructure the abstract making clear from the beginning that the investigations are performed among snow routines based on temperature-index methods only. - Line 34:

[Figure]

". . . often triggered by raising temperature". Is the main triggering source induced by air temperature or by incoming solar radiation, which is well represented by temperature? - Line 64-66: "Regarding the proportionality constant . . ." Is the constant catchment defined? Are there studies which take into account of the spatial variability (e.g. different altitude, topography, etc?) - Line 66-67: ".. one for temperature and another for net radiation". Doesn't this belong to the hybrid methods? - Line 115 Formula (3) Is T the daily average temperature? Some formulations take into account the cumulated temperature which exceeds the threshold, measured for example with 1 hour time step. Would these different formulation affect the results? - Section 2.2.1 Has the formula (5) been evaluated by using the available temperature data for the studied catchment? - Paragraph 2.2.5 What is the threshold used in the model as the maximum liquid water content retained in the pores (maximum water retention capacity)? - Section Results. I would plot the performance vs size of the catchment and altitude (also for a fixed configuration, given the high number of variable components).

---

## Referee Comment (RC3) · María José Polo (Referee) · 13 Mar 2020

General comments

This work analizes the performance of different snow routines based on the degree-day method in the framework of the HBV hydrological model. For this, runoff together with other snow-related variables are simulated in a large number of basins in Alpine areas in Central Europe and then compared to different sets of observations. The routines include different modifications for the snow routine components in HBV. Despite the significant variability found among cases, the results identified an exponential snowmelt function as the best modification in terms of model performance, followed by

the adoption of a seasonal degree-day factor; other processes, like refreezing, added little benefit to the model pointing out that complexity itself is not an advantage without careful model design. The work addresses an interesting topic for areas where physical modelling approaches demand larger data sets than the available observations, and it is very clearly presented. Despite the conclusions cannot be directly extrapolated to other snow regions in the world, the number of study cases cover a large area in Central Europe, where snow processes condition the hydrological response in many rivers.

I have some observations that can be assessed by the Authors to emphasize the applicability of the results and the scope of the study; some minor comments are also included.

1. The work includes all the different snow routines in the HBV model, and no other hydrological model is assessed. I suggest making it clear in the title that the assessment is done on the HBV performance, since "... for runoff modelling in mountainous areas in Central Europe", since it may lead to expect a wider scope of models. Additionally, some comments addressing whether the level of improvement or not obtained from each routine is affected by the model choice. At least, some reference to similar models should be included and some justification of what conclusions would be expected to be shared from simulations by other hydrological models.

2. A second issue is related to the spatial resolution of the input data, and potential scale effects. Gridded weather data in the Swiss cases, 1-km2 of gridded SWE, and 25-m cell size of the DEM, whereas point observations from stations and a 5-m DEM are used in the Czech catchments. Could you provide some assessment on these potential scale effects, and whether the source of weather data had an influence or not on the results? I also wonder whether using mean SWE values over each elevation zone, and point SWE measures, depending on the cases, could affect the results and comparison. Also, do you think that the results are scale-dependent of the cell size of the DEM used in the HBV model?

3. In the introduction, I miss some inclusions, like the importance of sublimation from the snow under certain conditions (not only in dry areas like we reported in Sierra Nevada-Spain, but also during the summer in the Alps and other regions, see Herrero and Polo, 2016), the existence of experimental catchments in the world devoted to snow processes research (see for example a recent Special Issue in Earth System Science Data on "Hydrometeorological data from mountain and alpine research catchments"), or the use of remote sensing sources to provide data to monitor snowpacks and snowmelt (many examples can be found, e.g. Dietz et al. 2012). Lines 55-60 should also address the limitations of degree-day approaches, and when they, although simple, are not an option.

4. I am curious about the performance of each routine regarding the snow cover distribution. Did you check also their ability to capture this by testing against some satellite images? This is very interesting in terms of model performance to identify the sources of improvement or not.

5. Since only four of the case studies were above 2000 m a.s.l. (only one above 2500 m), I think that some comment on how the results could change or not in higher elevation sites would shed light on their further applicability, especially in catchments where snowmelt is a higher fraction of runoff.

6. I fully agree with selecting just some examples to conduct the presentation of results. I think, however, that including more than just one catchment, and year, would add value to your results. You could suggest another one from a lower altitudinal range, coming from the Swiss area, so that the impact of the spatial scale effects could, if needed, also be discussed. It would be very nice being able to see selected results from all the cases, I would suggest their inclusion as supplement.

Other comments: 7. The gridded data of SWE in the Swiss cases were derived from a temperature-index model. Could this bias the performance of the routines?

8. Lines 259-260. Please, could you assess whether this decision could affect the

results or not.

9. Figure 4. Please, could you show also some validation results for this example case and year.

10. Lines 410-412. Any comment on why these different behaviours are found?

11. Lines 425-427. Reading this, I would conclude that runoff data/simulations are somehow limiting the model performance's improvement (see also your comments in lines 482-484, and in lines 496-499. Additionally, this content should be reflected in conclusions (lines 565-567), to be more specific.

12. I would suggest including some quantitative result in the conclusions, but I leave it up to the Authors.

I hope that these comments help the Authors to address further their results and can contribute to the final version of the manuscript.

References:

Dietz et al., 2012. International Journal of Remote SensingÂă33(13):4094-4134 https://doi.org/10.1080/01431161.2011.640964

Herrero and Polo, 2016. The Cryosphere, 10, 2981–2998, 2016 https://doi.org/10.5194/tc-10-2981-2016

Special Issue in Earth System Science Data on "Hydrometeorological data from mountain and alpine research catchments" https://www.earth-syst-sci-data.net/special_issue871.html

---

## Referee Comment (RC4) · Thomas Skaugen (Referee) · 13 Mar 2020

Review for "Complexity and performance of temperature-based snow routines for runoff modelling in mountainous areas in central Europe" by Lopez et al

General comments: This paper describes the testing of many alternative conceptual algorithms for snow modelling implemented in the Swedish HBV model. The suitability of the different algorithms has been assessed by split sample procedures for many catchments in Czechia and Switzerland. The paper is well written, well organized and the experimental setup seems, in principle, to be fine. However, the possible improvements of the tested alternative algorithms are extremely subtle and the authors

recommend exponential snowmelt function and seasonally varying degree-day factor based on tiny improvements which have not, as far as I can see, been tested for their significance. I think the objective of the paper is good, it would be nice if we in objective ways could agree on improved concepts in snowmodelling that when implemented would improve any model, but I am doubtful if the current methods are up for the task. The following issues need to be addressed in order to make paper suitable for publication.

1) The paper misses a major investigation on equifinality issues (see papers of K. Beven and J. Kirchner on this topic). The HBV model itself has a lot of freedom, i.e. parameters to be calibrated, and most of the suggested algorithms for possible improved snowmodelling add calibration parameters and hence to the problem of overparameterization. The point is that many of the suggested snow model modifications may have potential for being better at modelling snow, but the effect is impossible to isolate due to the overparameterization/equifinality. I have personal experience with trying to implement, what I thought was brilliant, ideas of improved snowmodelling to the HBV model. They were all insignificant, and after a while I realized that the compensating powers of all the parameters in HBV made it impossible to isolate and assess the effect of new algorithms (the frustration inspired the development of a new rainfall runoff model). The inclusion of the objective function for SWE is a step in the right direction, it narrows the freedom of the parameters, but probably not enough (you could try to also include Snow Covered Area, SCA). How many calibration parameters are there in the various model configurations? Are the numbers acceptable by any measure? Are their ranges physical at equifinality?

2) I would desire a more stringent terminology. Words like "efficient" and "complex" have really lost their true meaning in the literature of hydrological modelling. Effective parameters really mean parameters that lump many processes or represents areal averages and has little to do with efficiency. A non-linear formulation of a process is not necessarily complex if the parameters are physical and measurable. To me,

an overparameterized model where, due to the compensating behavior among the parameters, the degree-day factor is suddenly correlated to the parameter controlling the subsurface storage capacity is infinitely complex. Please consider rewriting the paragraph that starts at 525

3) There are several paragraphs subjectively praising the HBV model for its ability to simulate hydrological behavior for various catchment types (p. 24, l.476, p.25, l 520-25, p.26, l563). "Hydrology" is a wide term and comprises more than runoff (and SWE admittedly), what about the subsurface, SCA, evapotranspiration etc. How come we are just presented result for one catchment?

Specific comments:

P1, l.18, "popular" subjective

P1, l.27 "optimal degree of realism", rephrase

P2, l.143-44 How can "the limitations of data availability–" "pose a challenge to properly monitoring". rephrase. . ..

P2, l.45-46 "Furthermore. . ." This sentence does not relate to anything above.

P2, l.52 ..available at..

P2, l.53. ..in a distributed way.. Not always, see Skaugen et al., 2018 (Hydrology Research)

P2, l.58 ..relevant for.. Aren't they relevant everywhere?

P2, l.62 ..distribution function.. What is this?

P3, l.83-84. ..and investigated whether.. See major comment above.

P4, l.98. "well established", what does this mean? is it good or just old

P5, l.125. . .to it... Refers to HBV or the individual components

P5, l.125 is precipitation lapse rate missing in the table? Could we have all calibration parameters in the table?

P5, l.131. Heading, "Temperature and precipitation lapse rates"

P7, l.189. ..if somewhat.. How is it more realistic

P12, l.257. ..higher model complexity.. why more complex if you increase the temporal resolution

P12, l.257-58. "Other factors..", please elaborate

P12, l.266-67. Good, this fights the problem of overparameterization. Could even include SCA

P13, l.292. "efficiency", efficient how? Faster? gets more done? or just better?

P14, l.322..performance for...

P14, l.323-24. Can we accept improved performance for snow and decreased performance for runoff? I know that other authors have reported this, but is this not a clear indication of model structure flaw? Pleas elaborate, this is important

P18, l.373.. "catchment dependent.." I do not understand this sentence

P19, Figure. What does the y-axis represent, I struggle with this figure

P20, l.404-407. This paragraph is very complex, can you please explain better

P20, l.408 64 or 63 (see Table above)

P23, l.457 ..are dominant.. meaning strong or better?

P24, l.475. The first sentence is meaningless. Of course it is difficult to improve hydrological models, the processes are complex. The reason why it is difficult in the case of HBV could be due to the overparameterization, not because it has been widely used with acceptable results.
P24, l.487. .. runoff is modulated.. rephrase

P25, l.533. . . even if model complexity. . .in a sensible way.. The sentence is strange

P26, l.551 different settings.. please be more specific475.

P26, l.563. Unsubstantiated, we have only seen the result for one catchment.475.

P26, l.565. How to proceed with this "better approach", how to do it in practice?

References Skaugen, T., H. Luijting, T. Saloranta, D. Vikhamar-Schuler and K. Müller, 2018. In search of operational snow model structures for the future - comparing four snowmodels for 17 catchments in Norway. Hydrology Research, 49.6, https://doi.org/10.2166/nh.2018.198

---

## Author Comment (AC1) · 7 Apr 2020

**Authors' response to interactive comment by Reviewer #1 Juraj Parajka**

Black text: Reviewer comment

Blue text: Authors' response

We thank the reviewer for his valuable comments and suggestions that will help us improve our manuscript. Below we reply to each of these and explain how we will incorporate them into the manuscript.

The study evaluates snow and runoff performance of 64 snow routine alternatives based on degree-day approach in large sample of catchments (54) located in Swiss and Czech Republic. The snow routine variants are coupled with HBV conceptual hydrologic model and model simulations are evaluated in terms of observed daily runoff and snow water equivalent observations/estimates. The results indicate that exponential snowmelt function with no refreezing and seasonally variable degree-day factors are the most reliable/robust/accurate variants for snowmelt runoff simulations in selected catchments. Overall, this is an interesting study which is worth to publish. The topic is relevant and within the scope of the journal. The study is clearly written and has a good structure. The analyses and interpretations are based on larger sample of catchments which allows to draw interpretations/conclusions that are relevant for large region of similar physiographic conditions in Central Europe.

Thanks for these kind words.

I have only few comments/notes which can be considered (in my opinion) to add/extend/improve clarity and generality of findings. These include:

1) Perhaps it will be possible to refer here in general to variants of degree-day snow approach, not strictly limits the analysis to HBV variants. The results can be used/implemented in degree-day routines of different hydrological models. In this study, the variants are coupled with HBV concept of rainfall-runoff transformation, but I believe, at least the evaluation of snow efficiency is relevant to general degree-day approach.

We also agree that the evaluation of the snow simulations is relevant to the degree-day approach in general, beyond its use in the HBV model. Indeed, what we present here is a methodology to analyse the impact of using different alternative model structures for a specific purpose (here, snow processes) on the performance of a rainfall-runoff model over a large sample of catchments. We, of course, related the proposed modifications to the HBV model, as this is our tool to conduct the analysis. Nevertheless, the alternative model structures that we explored in this study are not only "HBV variants", but at the same time also variants of the degree-day approach as used in other hydrological models as well. We understand the concern of the reviewer and we will clarify in the manuscript that, while using the HBV model, our study also is an evaluation of the degree-day approach in general.

2) When coupling the 64 snow routine variants with HBV model, there is another interesting question, which can be discussed and this is the robustness/uncertainty of other HBV model

parameters. How consistent/different are the other HBV model parameters for different snow variants? Are, for example, field capacity or nonlinear runoff generation (beta) parameter values similar or compensating some effects of different snow routines?

The reviewer raises an important point here. In models such as the HBV model, model parameters can compensate each other, which makes the interpretation of any modifications rather challenging. While we did not include this in the manuscript, we performed a Monte Carlo sensitivity analysis on the HBV parameters. We provide figures resulting from these analyses for each catchment at the end of this comment (caption only provided for Figure 1). We found that, even if some of the variants (i.e. $T_{p,m}$, $\Delta P_e$, or $C_{0,s}$) produce compensating effects on some parameters (e.g. PCALT, FC, LP or BETA), this effect was only observed for some of the catchments. Overall, parameter values and sensitivity tended to be fairly consistent across all the tested model variants for most of the catchments. We are aware of these potential compensatory effects between model parameters which can mask the real impact of different snow routine variants and, therefore, decided to base the evaluation of the analysis on the ability of the different model variants to reproduce snow water equivalent in addition to stream runoff (which is how hydrological models are traditionally evaluated).

3) It is not clear which part of the snow accumulation/melt phases are described/evaluated by selected snow objective function? For some practical applications, for example, it will be interesting to see the difference in maximum snow water equivalent between the routines, or to what extent the model over or underestimates snow cover duration? To what extent are these aspects covered in current snow efficiency evaluation? Does a good simulation mean well represented maximum SWE or snow cover duration? Perhaps there are some differences in such efficiency between the variants.

Our evaluation of the performance of the snow simulations provides an overall assessment of the snow processes in the selected catchments. As the reviewer correctly states, the evaluation could also be based on more specific aspects such as magnitude or timing of maximum annual snow accumulation. The problem with measures based on specific aspects is that a perfect fit with regard to one single measure does not ensure a good overall performance (similar to individual flow indices or signatures in the case of runoff simulations, see Vis et al., 2015). This implies that a number of measures would be needed to be used together with the challenge to decide on appropriate ways to combine the measures into a single overall performance measure. While we agree that this could allow further assessments and might be valuable for future studies, we are afraid that such an additional analysis would be beyond the scope of this manuscript.

4) In our recent study (Sleziak et al., 2020) we found that there are quite significant differences in snow model performance (by using standard HBV degree-day approach) between lowland and alpine catchments in Austria. (Differences in terms of overestimation of snow cover in alpine and underestimation of snow cover in flatland catchments). Did you observe similar findings here?

Yes, in our study we also observe that model performance is generally lower for lowland catchments than for alpine ones. Our simulations also tended to underestimate snow water equivalent in lowland catchments. Regarding alpine catchments, we observed no clear pattern regarding over- or underestimation. Overall, it was quite frequent for the model to underestimate snow accumulation and delay the timing of the spring snowmelt season. We appreciate the comment of the reviewer and we will expand the related sections in our text and relate our results with the findings from the suggested article (Sleziak et al., 2020).

**Specific comments**

1) Abstract. Is the last sentence needed?

With this sentence we wanted to highlight some of the limitations of the results obtained in this study. However, as we also discuss these limitations in detail in the manuscript this "general disclaimer" might indeed not be needed. We will remove the last sentence from the abstract.

2) Introduction: It will be interesting to extend somewhat this section by referring to ways how can be/are degree-day routine parameters estimated in hydrological models.

We already do this to some extent in the methods section, where we present the different alternative model structures that we considered in this study, together with references. However, we understand that it would be relevant to refer to these different approaches and implementations in a general way in the introduction as well. We will, therefore, expand the introduction to give an overview of how different hydrological models use the degree-day method (see also response to major comment #1).

3) Data: How close are gridded snow water equivalent data to observations? Is there some bias related to the fact that this dataset is based on some type of degree-day model?

Using a temperature-index (TI) approach in both the runoff model as well as in the snow model providing validation data might indeed lead to some bias. Nevertheless, it has to be taken into account that the snow model makes use of a 3-dimensional sequential data assimilation (DA). The DA itself includes two methods which are both based on spatially correlated error statistics. For snow accumulation, an optimal interpolation approach uses the snow water equivalent station data to correct the simulated snowfall amounts. Concerning snowmelt, an ensemble Kalman filter updates snowmelt rates as well as liquid water content. Finally, the combination of both data assimilation approaches results in corrections of modelled snow water equivalent within all 1 by 1 km grid cells. Magnusson et al. (2014) investigate the performance in predicting snow water equivalent when using this DA approach and compare it to the TI model without DA. Based on 1033 samples from 45 stations, they show that using DA leads to improvements in predicting snow water equivalent.

4) Runoff model efficiency. Why only Nash-Sutcliffe based on logarithmic transformed discharges? It will be interesting to see also the model performance in terms of snowmelt runoff peaks.

When we designed the study, we gave some thought on potential evaluation metrics, among which were the NSE, MARE or snow cover fraction. Nevertheless, since the computational demands for conducting this study were considerable (large array of catchments and model modifications), we decided to just use two objective functions (one for snow processes and another for rainfall-runoff transformation) that would be as relevant as possible. We agree that testing the model performance in terms of snowmelt runoff peaks would be very interesting indeed. Looking at additional performance measures would be a valuable next step, but including this here would make the study overly complex due to the inclusion of too many aspects (see also answer to major comment #3).

5) P.15, l.355: Figure 3 or Figure 4?

This should indeed be Figure 4. We will correct the error.

6) Figure 4. Will it be possible to show such case for a year in the validation period?

The intention behind presenting only the calibration results was to keep a simple story and walk the reader through the results by adding complexity stepwise. Nevertheless, we understand that having some detailed validation results might add valuable information to the reader as well. We will modify the figure to include validation results for the same year.

7) Results: Will it be possible to present runoff and snow model efficiencies for each catchment in the Supplement?

This study includes many different catchments and model variants, in addition to two periods for cross-validation analysis using two different metrics and presenting all these data in a meaningful way is not easy. However, since we only presented absolute results for one catchment in the manuscript, we agree with the reviewer that it would be a good idea to, at least, provide summarised results from all the catchments, periods, and evaluation metrics in an appendix. We will, therefore, include a table showing median efficiency values for each catchment, and objective function, both for model calibration and validation for both periods for the default HBV model. Since we already included relative performance differences for all catchments and metrics in the manuscript (see Figure 5) we argue that this would be sufficient to provide a good overview of the model results.

Additionally, we could provide further figures similar to Figure 4 (including validation results, following the previous comment), for all catchments in an appendix as well. This appendix would, however, be rather extensive and we would appreciate the editor's guidance on whether to include this or not.

**References**

Sleziak et al. (2020) The effect of the snow weighting on the temporal stability of hydrologic model efficiency and parameters, Journal of Hydrology.

Magnusson, J., Gustafsson, D., Hüsler, F., & Jonas, T. (2014). Assimilation of point SWE data into a distributed snow cover model comparing two contrasting methods. *Water resources research*, *50*(10), 7816-7835.

Vis, M., Knight, R., Pool, S., Wolfe, W., & Seibert, J. (2015). Model calibration criteria for estimating ecological flow characteristics. *Water*, *7*(5), 2358-2381.

**Sensitivity analysis**

[Figure]

*Figure 1. Sensitivity analysis on parameters other than those from the snow routine for a catchment CH-101. Each subplot shows the model performance (y-axis) as a function of the values of a single free parameter (x-axis) with all other parameters set at the median best value of all 10 calibration trials for each of the single modifications to the snow routine of the model (see Table 1 in the manuscript). These results shown here are for model calibration on period 1.*

[Figure]

[Figure]

[Figure]

[Figure]

[Figure]

[Figure]

CH-202

CH-203

CH-204

CH-205

[Figure]

CH-302

CH-303

[Figure]

CZ-103

[Figure]

[Figure]

CZ-206

[Figure]

CZ-207

[Figure]

[Figure]

[Figure]

CZ-301

CZ-302

[Figure]

[Figure]

[Figure]

[Figure]

CZ-406

[Figure]

CZ-407

---

## Author Comment (AC3) · 7 Apr 2020

The comment was uploaded in the form of a supplement:
https://www.hydrol-earth-syst-sci-discuss.net/hess-2020-57/hess-2020-57-AC3-supplement.pdf

---

## Author Comment (AC4) · 7 Apr 2020

**Authors' response to interactive comment by Reviewer #4 Thomas Skaugen**

Black text: Reviewer comment

Blue text: Authors' response

We thank the reviewer for his valuable comments and suggestions to improve our contribution. Below we reply to each of them and explain how we will incorporate them into the manuscript.

**General comments**

This paper describes the testing of many alternative conceptual algorithms for snow modelling implemented in the Swedish HBV model. The suitability of the different algorithms has been assessed by split sample procedures for many catchments in Czechia and Switzerland. The paper is well written, well organized and the experimental setup seems, in principle, to be fine. However, the possible improvements of the tested alternative algorithms are extremely subtle and the authors recommend exponential snowmelt function and seasonally varying degree-day factor based on tiny improvements which have not, as far as I can see, been tested for their significance. I think the objective of the paper is good, it would be nice if we in objective ways could agree on improved concepts in snow modelling that when implemented would improve any model, but I am doubtful if the current methods are up for the task. The following issues need to be addressed in order to make paper suitable for publication.

Yes, the effects of the different modifications are small. Nevertheless, even if the average model performance improvement for the recommended model modifications are small on average, these modifications are significant for individual catchments, and do not lead to decreased model performance in any case. Minor changes had to be expected in general because we use a large sample of catchments in our study. Any improvements will, thus, tend to average out and look less impressive. We argue that while these improvements might indeed be small, our evaluation based on many catchments means that the findings are more robust than in many previous studies.

1) The paper misses a major investigation on equifinality issues (see papers of K. Beven and J. Kirchner on this topic). The HBV model itself has a lot of freedom, i.e. parameters to be calibrated, and most of the suggested algorithms for possible improved snow modelling add calibration parameters and hence to the problem of overparameterization. The point is that many of the suggested snow model modifications may have potential for being better at modelling snow, but the effect is impossible to isolate due to the overparameterization/equifinality. I have personal experience with trying to implement, what I thought was brilliant, ideas of improved snow modelling to the HBV model. They were all insignificant, and after a while I realized that the compensating powers of all the parameters in HBV made it impossible to isolate and assess the effect of new algorithms (the frustration inspired the development of a new rainfall runoff model). The inclusion of the objective function for SWE is a step in the right direction, it narrows the freedom of the parameters, but probably not enough (you could try to also include Snow Covered Area, SCA). How many calibration parameters are there in the various model configurations? Are the numbers acceptable by any measure? Are their ranges physical at equifinality?

Indeed, equifinality is an issue in many hydrological models, and HBV is no exception to this. However, compared to many other models, the HBV model uses rather few parameters and parameter uncertainty is thus, smaller. The particular version used here, HBV-light, has been frequently used to address parameter uncertainty in the past years. So, while parameter uncertainty is an issue, we argue that we in the past have gathered quite some experience related to this issue. That being said, and while we did not include this in the manuscript, we performed a Monte Carlo sensitivity analysis on the HBV parameters. We attach some figures resulting from these analyses at the end of the comment (a caption is only provided for Figure 1). We found that, even if some of the variants (i.e. $T_{p,m}$, $\Delta P_e$, or $C_{0,s}$) produce compensating effects on some parameters (e.g. PCALT, FC, LP or BETA), this effect was only observed for some of the catchments. Overall, parameter values and sensitivity tended to be fairly consistent across all the tested model variants for most of the catchments. It is also important to note that, even if some parameter profiles might look quite flat in these figures, hinting to equifinality issues, in most cases this is an artefact form forcing the same y-axis range across all subplots, which makes some of the parameter shapes difficult to appreciate. As we previously mentioned, we took consideration of both potential equifinality and parameter compensation issues in our analysis. We will emphasise this in the revised manuscript.

Most of the modifications that we tested in this study add only one extra parameter to the snow routine of the model (which consists of 5 parameters: degree-day factor, refreezing coefficient, threshold temperature, water holding capacity of the snowpack, and snowfall correction factor). Additionally, as the reviewer noticed, we assessed the impact of these modifications on the output of the snow routine (i.e., snow water equivalent) to avoid interactions from the other model routines and parameters. We also tested the use of other snow-related objective functions such as snow cover fraction. However, in the end, we decided not to use this measure because snow cover fraction does not provide a direct estimation of the amount of freshwater stored in the snow, which makes this parameter difficult to relate to the mass-balance approach of HBV. Additionally, cloud cover was an issue in the tests we performed. Furthermore, using this objective function could lead to large overestimations of snow water equivalent from, for instance, light snowfall events in late spring, when most of the catchment is no longer snow-covered but when there is still a significant storage of snow at high elevations. Such events could make the snow cover fraction jump up to 100% while the actual catchment-wide snow water equivalent would only have marginally increased. Finally, the scope of the study, including a large number of catchments and model alternatives, meant a large computational demand. We, therefore, made an effort to identify the most relevant metrics for evaluating the model for both snow processes (i.e. Rw) and rainfall-runoff transformation (i.e. Rln(Q)).

Regarding the decision to also evaluate the results respect to stream runoff, it was taken based on the common use of many hydrological models. We know that HBV is an imperfect hydrological model (as is the case for all models) based on certain assumptions that lead to issues such as parameter compensation. Even so, since models based on these assumptions will continue to be used in the foreseeable future mostly for runoff simulation, we wanted to ensure that the modifications we introduced to the model would produce acceptable results within the imperfect framework of the model. We agree with the reviewer in that better modelling approaches need to be found, but we also think that the available tools need to be evaluated and improved upon as well.

Coming back to the number of additional calibration parameters for the various model configurations, this number varies between 1 and 3 parameters for single modifications. Several of the modifications that we selected were derived from observations, such as the seasonally-variable temperature lapse rate (Rolland, 2002) or the exponential precipitation phase partition (Magnusson,

2014). For the other parameters, we used constrained ranges (based on, for instance, Seibert (1999) for the default HBV structure) to ensure that parameter values do not become unrealistic. When different modifications are used simultaneously, the number may go up to 9 parameters. We regard this number as excessive and a clear over-parameterisation, which opposes the aim of preserving a simple structure with as few parameters as possible. However, we still included this variant in the evaluation for the sake of completeness. We will clarify this in the revision.

2) I would desire a more stringent terminology. Words like "efficient" and "complex" have really lost their true meaning in the literature of hydrological modelling. Effective parameters really mean parameters that lump many processes or represents areal averages and has little to do with efficiency. A non-linear formulation of a process is not necessarily complex if the parameters are physical and measurable. To me, an over-parameterized model where, due to the compensating behavior among the parameters, the degree-day factor is suddenly correlated to the parameter controlling the subsurface storage capacity is infinitely complex. Please consider rewriting the paragraph that starts at 525

We understand the concerns of the reviewer and agree with the need for a clear and concise terminology. Ideally, model efficiency relates to the definition the reviewer provides here. Nevertheless, as the reviewer also mentions, these terms can also be used to refer to other concepts such as models that provide acceptable results, even if for the wrong reasons. As explained in reply to the previous comment, in this study, we aim for both improving the quality of the processes conceptualisation in the model and ensuring that this improved conceptualisation works well with the imperfect nature of the model. That being said, we will revise the manuscript to ensure that the terminology we use is appropriate and concise, and we will rewrite the aforementioned paragraph to iron out inaccurate references to complexity and efficiency.

3) There are several paragraphs subjectively praising the HBV model for its ability to simulate hydrological behavior for various catchment types (p. 24, l.476, p.25, l 520-25, p.26, l563). "Hydrology" is a wide term and comprises more than runoff (and SWE admittedly), what about the subsurface, SCA, evapotranspiration etc. How come we are just presented result for one catchment?

The reviewer makes an important remark here to the need for objectively assessing the strengths and weaknesses of the selected models and design choices. We will revise the passages mentioned by the reviewer to ensure that the demands for objectivity are met adequately.

Regarding the term "hydrology", we agree with the reviewer in that hydrology is a wide term and that there are many relevant variables and processes that often get overlooked in favour of – in most cases – stream runoff. We refer to the reply to major comment 1 for an explanation on the variables we used to assess the analysis presented in this contribution. Similar to the previous point, we will, of course, consider this comment when revising the manuscript and we will replace the generic references to "hydrology" by more concise terms. Additionally, we will specify that we do not investigate processes other than snowpack accumulation and evolution and its impact on stream runoff (i.e. no subsurface, chemical... processes).

We only presented results for a single catchment to not make the paper excessively long or complex. The analysis we present here includes a large array of catchments and alternative model variants, which make it impractical to present all the results in a detailed way without making the manuscript overly cumbersome. Nevertheless, we understand the concerns of the reviewer, and we will, therefore, add an appendix with the default HBV model performance values for all catchments, objective functions, and calibration/validation efforts included in this study. We believe that, based

on this table and on the figures included in the manuscript, the reader will have all the necessary information to evaluate this contribution. Additionally, if the editor and reviewer find this useful, we could provide detailed figures similar to Figure 4 (but including model validation results) for all catchments in the study in another appendix (54 figures). We provide these figures at the end of this response for the reviewer to evaluate (caption only provided for Figure 55).

**Specific comments**

P1, l.18, "popular" subjective

We agree with the reviewer that this is a subjective expression. We will rephrase the text to emphasise that this model has been (and still is) widely used in many different settings.

P1, l.27 "optimal degree of realism", rephrase

We will rephrase the expression based on the responses to the major comments above.

P2, l.143-44 How can "the limitations of data availability–" "pose a challenge to properly monitoring". Rephrase....

With this sentence we wanted to express that monitoring hydrological processes (and more specifically snow processes in this case) is challenging with limited observations. We will rephrase the sentence for clarification.

P2, l.45-46 "Furthermore..." This sentence does not relate to anything above.

The idea we wanted to express with the final part of this paragraph was that, if having (limited) observations already makes it challenging to properly assess the current evolution of the hydrological processes (in this case snow processes), to be able to predict their future evolution with – obviously – no observations on the potential changes is even more complicated. We will rephrase this sentence to make it fit better in the paragraph.

P2, l.52 ..available at..

We will correct this mistake.

P2, l.53. ..in a distributed way.. Not always, see Skaugen et al., 2018 (Hydrology Research)

We admit that we over-generalised the identification of energy-based approaches with distributed hydrological models. We will address this by explicitly mentioning lumped and semi-distributed models that are based on these approaches. We thank the reviewer for providing this reference.

P2, l.58 ..relevant for.. Aren't they relevant everywhere?

Indeed, using radiation data in addition to temperature data is relevant everywhere. Nevertheless, the benefits of using these approaches are most notable in the catchments described in the aforementioned sentence than in other types of catchments, where the impact is more modest. We did test such an approach for this set of catchments and found that the improvements were small while requiring an additional data source and calibration parameter.

P2, l.62 ..distribution function.. What is this?

We intended to say "bucket-type model" / "conceptual model". We will correct this error in the revision.

P3, l.83-84. ..and investigated whether.. See major comment above.

We refer to the reply to major comment 1 and 2 above. We will revise the text to ensure that the terminology we used is concise and relevant.

P4, l.98. "well established", what does this mean? is it good or just old

We would argue that it means a bit of both. Indeed, the degree-day approach is an old conceptualisation of snowmelt processes, that has been in use for a long time already, and it has been evaluated, tested, and implemented in many different studies and models due to its low data requirements and explanatory power. We will rephrase this expression to include these nuances.

P5, l.125...to it... Refers to HBV or the individual components

It refers to the snow routine of HBV. We will reformulate this to avoid confusions.

P5, l.125 is precipitation lapse rate missing in the table? Could we have all calibration parameters in the table?

We intended this table to present the proposed modifications to the snow routine of HBV so, since we decided not to test any alternative to the precipitation lapse-rate, we did not include it in this table. Nevertheless, we understand the confusion of the reviewer, and we will, therefore, add it to the table.

Regarding the calibration parameters, we argue that including all calibration parameters is out of the scope of this table. We will instead add a sentence in Section 2.1 (in which the individual parameters are described) summarising the number of calibration parameters in the snow routine of the model. The reader will then be able to easily calculate the number of calibration parameters that are needed for each model variant.

P5, l.131. Heading, "Temperature and precipitation lapse rates"

We will modify the section header (see also the previous comment).

P7, l.189. ..if somewhat.. How is it more realistic

It is more realistic than the one used in HBV since it does not have an abrupt transition (i.e. snowmelt being 0 up to a threshold and increasing linearly thereafter) in the change of snowmelt rate. Nevertheless, it does require the use of an additional parameter to control for the smoothness of the snowmelt transition.

P12, l.257. ..higher model complexity.. why more complex if you increase the temporal resolution

We explain this in the following sentences. We argue that to represent these processes at a sub-diurnal time step correctly, we would need to include additional parameters to control for processes that become relevant at these resolutions. Nevertheless, "complex" might not be the appropriate term here; "detailed" might be more suitable in this context (see also the reply to major comment 2 above). We will rephrase this sentence to be more concise in the terminology.

P12, l.257-58. "Other factors..", please elaborate

With this sentence, we meant that factors such as the transport time of meltwater from the snowpack to the stream become relevant for sub-daily time steps.

P12, l.266-67. Good, this fights the problem of over-parameterization. Could even include SCA

We refer to the reply to major comment 1 for a discussion on why we finally did not use snow cover fraction as a metric to evaluate this study.

P13, l.292. "efficiency", efficient how? Faster? gets more done? or just better?

In here, we referred to model performance when evaluating the model against each of the chosen objective functions. As already mentioned in major comment 2 above, we will revise the manuscript to ensure that the terminology is concise and relevant throughout the text.

P14, l.322..performance for...

We will correct the error.

P14, l.323-24. Can we accept improved performance for snow and decreased performance for runoff? I know that other authors have reported this, but is this not a clear indication of model structure flaw? Please elaborate, this is important

The reviewer raises an important point here, which is linked to the major comment above. Indeed, this kind of observations should make the hydrological community think about the complexity issues and limitations of the current generation of models and use this evidence to guide further research efforts that allow us to increase our understanding of these processes (including their connexions and feedback mechanisms) as well as to design and implement better (and usable) modelling strategies that avoid these issues. Nevertheless, this issue is beyond the scope of this manuscript, which attempts to improve an existing, yet imperfect tool by exploring, testing, and evaluating the suitability of existing alternative structures.

P18, l.373.. "catchment dependent.." I do not understand this sentence

With this sentence, we wanted to express that there are some modifications which have a clear and consistent impact on most catchments (either negative or positive), while other modifications produce either positive or negative impacts depending on the catchment. We will rephrase the sentence to make it easier to understand.

P19, Figure. What does the y-axis represent, I struggle with this figure

The Y-axis shows the rank spread of each modification across all the catchments in the study, and each column adds up to 100%. So, for a given modification it shows for which percentage of catchments it is the best model structure, the second-best model structure, and so on. So, for instance, for the top left subplot, using a seasonal degree-day factor is the best alternative (among all the single modifications + default HBV structure) for ~80% of the tested catchments, the second-best alternative for ~10% of the catchments, and so on.

P20, l.404-407. This paragraph is very complex, can you please explain better

This paragraph is intended to clarify why, in the case of introducing 5 modifications to the snow routine (i.e. modifying each of the snow routine components that we evaluated), there are only three possible alternatives. The only available alternative representations (see also Table 1) are used for the lapse rate (i.e. $\Gamma_s$), the threshold temperature (i.e. $T_{P,M}$), the degree-day factor (i.e. $C_{0,s}$), and snowmelt and refreezing (i.e. $M_e$), in combination with one of the three alternative representations for the precipitation phase partition (i.e. $\Delta P_l$, $\Delta P_s$ or $\Delta P_e$). We will rephrase the paragraph to make it easier to understand.

P20, l.408 64 or 63 (see Table above)

We considered 64 different model structures, including the default HBV structure. In Table 3 we only showed modifications to the default HBV structure. Nevertheless, we understand that this can lead to confusions and will change the table accordingly.

P23, l.457 ..are dominant.. meaning strong or better?

In here, by "dominant" we meant the modifications to single components that appear most frequently in the top-ranking model variants. We will rephrase this to make it more concise.

P24, l.475. The first sentence is meaningless. Of course it is difficult to improve hydrological models, the processes are complex. The reason why it is difficult in the case of HBV could be due to the over-parameterization, not because it has been widely used with acceptable results.

This sentence was meant as an introduction to the discussion so, taking into account the major comments by the reviewer, we will modify just to state that we observed that it is difficult to improve hydrological models like HBV. In the discussion, we will delve deeper into the reasons why such models are difficult to improve.

P24, l.487. .. runoff is modulated.. Rephrase

The intention with this sentence was to stress that, as already pointed out before, the final model output is the result of the interaction between the different routines of the model. This, as the reviewer points out in another comment, may be related to compensating effects between parameters but also to a loss of the signal from any modifications made on the snow routine. It is, therefore, to be expected that efficiency changes are minor when evaluating the model based on this variable. We will rephrase the sentence taking the previous discussion into account.

P25, l.533...even if model complexity...in a sensible way.. The sentence is strange

The intention with this sentence was to point out that, if there were enough data available and knowledge about the processes that to be simulated, then it would be justified to add more complexity (here understood as a more detailed description of the processes) to the model. We will rephrase the sentence to make it clearer.

P26, l.551 different settings.. please be more specific.

By different settings, we were referring to the geological, geographic, climatological, and hydrological characteristics that define the hydrological behaviour of a given catchment. We will make this sentence more specific.

P26, l.563. Unsubstantiated, we have only seen the result for one catchment.

We understand the concern of the reviewer regarding drawing general conclusions about the whole study when we only presented the details of a single catchment in the manuscript. We took this decision to facilitate the storyline of the manuscript and not overwhelm the reader with endless results. We refer to major comment #3 for the relevant modifications concerning this comment that we will include in the revision. We hope that the aforementioned changes provide enough evidence to support this conclusion.

P26, l.565. How to proceed with this "better approach", how to do it in practice?

In this conclusion point, we state that carefully assessing which objective, the necessary level of detail (see previous comment on complexity vs detail), and data availability to each case is a better approach than just picking whichever model we are familiar with or have a preference for.

Obviously, this is easier said than done but, in this contribution, we aim to provide a methodology to do such an assessment over a large sample of catchments and model structure variants for a specific purpose.

References

Magnusson, J., Gustafsson, D., Hüsler, F., & Jonas, T. (2014). Assimilation of point SWE data into a distributed snow cover model comparing two contrasting methods. *Water resources research*, *50*(10), 7816-7835.

Rolland, C. (2003). Spatial and seasonal variations of air temperature lapse rates in Alpine regions. *Journal of climate*, *16*(7), 1032-1046.

Seibert, J. (1999). Regionalisation of parameters for a conceptual rainfall-runoff model. *Agricultural and forest meteorology*, *98*, 279-293.

Skaugen, T., H. Luijting, T. Saloranta, D. Vikhamar-Schuler and K. Müller,2018. In search of operational snow model structures for the future - comparing four snow models for 17 catchments in Norway. Hydrology Research, 49.6, https://doi.org/10.2166/nh.2018.198

**Sensitivity analysis**

[Figure]

*Figure 1. Sensitivity analysis on parameters other than those from the snow routine for a catchment CH-101. Each subplot shows the model performance (y-axis) as a function of the values of a single free parameter (x-axis) with all other parameters set at the median best value of all 10 calibration trials for each of the single modifications to the snow routine of the model (see Table 1 in the manuscript). These results shown here are for model calibration on period 1.*

[Figure]

[Figure]

[Figure]

[Figure]

[Figure]

[Figure]

CH-204

CH-205

[Figure]

CH-302

CH-303

**CZ-103**

[Figure]

**CZ-104**

[Figure]

[Figure]

[Figure]

[Figure]

[Figure]

[Figure]

CZ-301

CZ-302

[Figure]

[Figure]

[Figure]

[Figure]

**Hydrographs**

[Figure]

*Figure 55 Time series (October 2003 – September 2004) for the catchment CH-101. Top: daily mean air temperature and total precipitation. Middle: model calibration results. Bottom: model validation results. The model calibration and validation are further subdivided into (top) catchment-average observed (grey line) and simulated snow water equivalent (HBV in blue and the model structure modification including a seasonally-varying degree-day factor, C0,s in orange), and (bottom) observed (grey line) and simulated stream runoff (HBV in blue and the model structure modification including a seasonal degree-day factor in orange). The grey field represents the period used when calibrating the model against the logarithmic stream runoff. The uncertainty fields for model*

[Figure]

[Figure]

[Figure]

[Figure]

[Figure]

[Figure]

[Figure]

[Figure]

[Figure]

[Figure]

[Figure]

[Figure]

[Figure]

[Figure]

[Figure]

[Figure]

[Figure]

[Figure]

[Figure]

[Figure]

[Figure]

[Figure]

[Figure]

[Figure]

[Figure]

[Figure]

[Figure]

[Figure]

[Figure]

[Figure]

[Figure]

[Figure]

[Figure]

[Figure]

[Figure]

[Figure]

[Figure]

[Figure]

[Figure]

[Figure]

[Figure]

[Figure]

[Figure]

[Figure]

[Figure]

[Figure]

[Figure]

[Figure]

[Figure]

[Figure]

[Figure]

[Figure]

[Figure]

---

## Author Response (AR1)

Dear HESS Topical Editor Elena Toth,

Thank you for your efforts with our manuscript. We have now addressed the issues raised by the reviewers and included most of their comments and suggestions. Below we provide a point-by-point reply to each of the comments (blue text) as well as an explanation on how we included them in the text or, in the case where we did not include them, why we did not do so (red text). Among other changes, we adapted the title of the manuscript to better reflect the significance of this contribution. Additionally, we made general language improvements to the manuscript to improve its readability.

Thereafter we provide a marked-up version of the revised manuscript so that all the modifications from the previous version can be tracked. In this revised version we included an appendix containing a table of HBV model performance values for all catchments. Additionally, in a separate file we provide a supplement containing selected results (both for model calibration and validation) for each catchment.

With the aforementioned modifications and additions to the original manuscript we hope that this contribution meets the quality requirements to be published at Hydrology and Earth System Sciences.

Kind regards,

Marc Girons Lopez, Marc Vis, Michal Jenicek, Nena Griessinger and Jan Seibert

**Authors' response to interactive comment by Reviewer #1 Juraj Parajka**

We thank the reviewer for his valuable comments and suggestions that will help us improve our manuscript. Below we reply to each of these and explain how we incorporated them into the manuscript.

The study evaluates snow and runoff performance of 64 snow routine alternatives based on degree-day approach in large sample of catchments (54) located in Swiss and Czech Republic. The snow routine variants are coupled with HBV conceptual hydrologic model and model simulations are evaluated in terms of observed daily runoff and snow water equivalent observations/estimates. The results indicate that exponential snowmelt function with no refreezing and seasonally variable degree-day factors are the most reliable/robust/accurate variants for snowmelt runoff simulations in selected catchments. Overall, this is an interesting study which is worth to publish. The topic is relevant and within the scope of the journal. The study is clearly written and has a good structure. The analyses and interpretations are based on larger sample of catchments which allows to draw interpretations/conclusions that are relevant for large region of similar physiographic conditions in Central Europe.

Thanks for these kind words.

I have only few comments/notes which can be considered (in my opinion) to add/extend/improve clarity and generality of findings. These include:

1) Perhaps it will be possible to refer here in general to variants of degree-day snow approach, not strictly limits the analysis to HBV variants. The results can be used/implemented in degree-day routines of different hydrological models. In this study, the variants are coupled with HBV concept of rainfall-runoff transformation, but I believe, at least the evaluation of snow efficiency is relevant to general degree-day approach.

We also agree that the evaluation of the snow simulations is relevant to the degree-day approach in general, beyond its use in the HBV model. Indeed, what we present here is a methodology to analyse the impact of using different alternative model structures for a specific purpose (here, snow processes) on the performance of a rainfall-runoff model over a large sample of catchments. We, of course, related the proposed modifications to the HBV model, as this is our tool to conduct the analysis. Nevertheless, the alternative model structures that we explored in this study are not only "HBV variants", but at the same time also variants of the degree-day approach as used in other hydrological models as well.

We adapted the manuscript in general and more specifically the introduction to emphasise that our study is an evaluation of the degree-day approach in general which uses the HBV model as a tool to conduct the investigation.

2) When coupling the 64 snow routine variants with HBV model, there is another interesting question, which can be discussed and this is the robustness/uncertainty of other HBV model parameters. How consistent/different are the other HBV model parameters for different snow variants? Are, for example, field capacity or nonlinear runoff generation (beta) parameter values similar or compensating some effects of different snow routines?

The reviewer raises an important point here. In models such as the HBV model, model parameters can compensate each other, which makes the interpretation of any modifications rather challenging. While we did not include this in the manuscript, we performed a Monte Carlo sensitivity analysis on the HBV parameters. We provide figures resulting from these analyses for each catchment at the end of this comment (caption only provided for Figure 1). We found that, even if some of the variants (i.e. $T_{p,m}$, $\Delta P_e$, or $C_{0,s}$) produce compensating effects on some parameters (e.g. PCALT, FC, LP or BETA), this effect was only observed for some of the catchments. Overall, parameter values and sensitivity tended to be fairly consistent across all the tested model variants for most of the catchments. We are aware of these potential compensatory effects between model parameters which can mask the real impact of different snow routine variants and, therefore, decided to base the evaluation of the analysis on the ability of the different model variants to reproduce snow water equivalent in addition to stream runoff (which is how hydrological models are traditionally evaluated).

In the revised version of the manuscript, we mentioned and discussed the sensitivity analysis both in the methods, results, and discussion sections (paragraphs starting at lines 329, 508, and 572) but we did not include any further figures in order to not make the manuscript even more complex.

3) It is not clear which part of the snow accumulation/melt phases are described/evaluated by selected snow objective function? For some practical applications, for example, it will be interesting to see the difference in maximum snow water equivalent between the routines, or to what extent the model over or underestimates snow cover duration? To what extent are these aspects covered in current snow efficiency evaluation? Does a good simulation mean well represented maximum SWE or snow cover duration? Perhaps there are some differences in such efficiency between the variants.

Our evaluation of the performance of the snow simulations provides an overall assessment of the snow processes in the selected catchments. As the reviewer correctly states, the evaluation could also be based on more specific aspects such as magnitude or timing of maximum annual snow accumulation. The problem with measures based

on specific aspects is that a perfect fit with regard to one single measure does not ensure a good overall performance (similar to individual flow indices or signatures in the case of runoff simulations, see Vis et al., 2015). This implies that a number of measures would be needed to be used together with the challenge to decide on appropriate ways to combine the measures into a single overall performance measure.

While we agree that this could allow further assessments and might be valuable for future studies, we are afraid that such an additional analysis would be beyond the scope of this manuscript.

4) In our recent study (Sleziak et al., 2020) we found that there are quite significant differences in snow model performance (by using standard HBV degree-day approach) between lowland and alpine catchments in Austria. (Differences in terms of overestimation of snow cover in alpine and underestimation of snow cover in flatland catchments). Did you observe similar findings here?

Yes, in our study we also observe that model performance is generally lower for lowland catchments than for alpine ones. Our simulations also tended to underestimate snow water equivalent in lowland catchments. Regarding alpine catchments, we observed no clear pattern regarding over- or underestimation. Overall, it was quite frequent for the model to underestimate snow accumulation and delay the timing of the spring snowmelt season.

We appreciate the comment of the reviewer. We expanded the discussion of the manuscript and relate our results with the findings from the suggested article (Sleziak et al., 2020) (paragraph starting at line 547).

**Specific comments**

1) Abstract. Is the last sentence needed?

With this sentence we wanted to highlight some of the limitations of the results obtained in this study. However, as we also discuss these limitations in detail in the manuscript this "general disclaimer" might indeed not be needed.

We removed the last sentence from the abstract.

2) Introduction: It will be interesting to extend somewhat this section by referring to ways how can be/are degree-day routine parameters estimated in hydrological models.

We already do this to some extent in the methods section, where we present the different alternative model structures that we considered in this study, together with references. However, we understand that it would be relevant to refer to these different approaches and implementations in a general way in the introduction as well.

We expanded the introduction to give an overview of how different hydrological models use the degree-day method (paragraph starting at line 70) (see also response to major comment #1).

3) Data: How close are gridded snow water equivalent data to observations? Is there some bias related to the fact that this dataset is based on some type of degree-day model?

Using a temperature-index (TI) approach in both the runoff model as well as in the snow model providing validation data might indeed lead to some bias. Nevertheless, it has to be taken into account that the snow model makes use of a 3-dimensional sequential data assimilation (DA). The DA itself includes two methods which are both based on spatially correlated error statistics. For snow accumulation, an optimal interpolation approach uses the snow water equivalent station data to correct the simulated snowfall amounts. Concerning snowmelt, an ensemble Kalman filter updates snowmelt rates as well as liquid water content. Finally, the combination of both data assimilation approaches results in corrections of modelled snow water equivalent within all 1 by 1 km grid cells. Magnusson et al. (2014) investigate the performance in predicting snow water equivalent when using this DA approach and compare it to the TI model without DA. Based on 1033 samples from 45 stations, they show that using DA leads to improvements in predicting snow water equivalent.

We included a brief discussion on the possible impact of using a degree-day model for both the hydrological model and the estimation of snow water equivalent for Swiss catchments in the methods section (paragraph starting at line 244).

4) Runoff model efficiency. Why only Nash-Sutcliffe based on logarithmic transformed discharges? It will be interesting to see also the model performance in terms of snowmelt runoff peaks.

When we designed the study, we gave some thought on potential evaluation metrics, among which were the NSE, MARE or snow cover fraction. Nevertheless, since the computational demands for conducting this study were considerable (large array of catchments and model modifications), we decided to just use two objective functions (one for snow processes and another for rainfall-runoff transformation) that would be as relevant as possible. We agree that testing the model performance in terms of snowmelt runoff peaks would be very interesting indeed.

Looking at additional performance measures would be a valuable next step, but including this here would make the study overly complex due to the inclusion of too many aspects (see also answer to major comment #3).

5) P.15, l.355: Figure 3 or Figure 4?

This should indeed be Figure 4.

We corrected the error.

6) Figure 4. Will it be possible to show such case for a year in the validation period?

The intention behind presenting only the calibration results was to keep a simple story and walk the reader through the results by adding complexity stepwise. Nevertheless, we understand that having some detailed validation results might add valuable information to the reader as well.

135 We modified the figure to include validation results for the same year.

7) Results: Will it be possible to present runoff and snow model efficiencies for each catchment in the Supplement?

This study includes many different catchments and model variants, in addition to two periods for cross-validation analysis using two different metrics and presenting all these data in a meaningful way is not easy. However, since

140 we only presented absolute results for one catchment in the manuscript, we agree with the reviewer that it would be a good idea to, at least, provide summarised results from all the catchments, periods, and evaluation metrics in an appendix.

In an appendix to the manuscript, we included a table showing median efficiency values for each catchment, and objective function, both for model calibration and validation for both periods for the default HBV model.

145 Additionally, we included a supplement containing further figures similar to Figure 4 (including validation results, following the previous comment), for all catchments.

**References**

Sleziak et al. (2020) The effect of the snow weighting on the temporal stability of hydrologic model efficiency and

150 parameters, Journal of Hydrology.

Magnusson, J., Gustafsson, D., Hüsler, F., & Jonas, T. (2014). Assimilation of point SWE data into a distributed snow cover model comparing two contrasting methods. *Water resources research*, *50*(10), 7816-7835.

Vis, M., Knight, R., Pool, S., Wolfe, W., & Seibert, J. (2015). Model calibration criteria for estimating ecological flow characteristics. *Water*, *7*(5), 2358-2381.

155

**Authors' response to interactive comment by Reviewer #2 Giacomo Bertoldi**

In attach some specific comments of Valentina Premier Ph.D. working with me.

We thank the reviewer for her valuable comments and suggestions that will help us improve our manuscript.

160   Below we reply to each of them and explain how we will incorporate them into the manuscript.

The paper applies some modifications on the snow routine of the HBV model. Main results are that an increasing complexity does not lead to increasing performance. The most positively influencing modification is the use of an exponential snowmelt function and of a seasonally variable degree-day factor.

Some comments follow:

165   - Line 15-17: "However, [...] support tool" This sentence is not really clear to me. In general, I would restructure the abstract making clear from the beginning that the investigations are performed among snow routines based on temperature-index methods only.

With this sentence we wanted to point out that the implications of the decisions on which model structure to use for a given application are not always adequately addressed.

170   We rephrase this sentence to clarify this point. Additionally, we revised the abstract to clarify that the study builds on temperature-index methods.

- Line 34: "... often triggered by raising temperature". Is the main triggering source induced by air temperature or by incoming solar radiation, which is well represented by temperature?

Incoming solar radiation is indeed an important driver behind snowmelt, perhaps the most important one for

175   open areas. This parameter is also strongly correlated with air temperature. Nevertheless, there are fluctuations in temperature that cannot be explained by incoming solar radiation alone, but by other processes such as lateral energy transfers, among others. For instance, snow also melts in locations with very little direct sunlight by the effect of temperature alone, such as under the canopy.

We clarified the text to make it more specific in respect to this, mentioning the important contribution of

180   incoming solar radiation and its correlation with temperature (paragraph starting at line 32).

- Line 64-66: "Regarding the proportionality constant ..." Is the constant catchment defined? Are there studies which take into account of the spatial variability (e.g. different altitude, topography, etc?)

Yes, since we only use a single vegetation zone per catchment (see paragraph starting at line 117), the proportionality constant is catchment-defined. By defining different vegetation zones this parameter could take into account e.g. aspect, forested areas vs bare ground, etc. This would however come at the cost of having additional free parameters for calibration and make this study overly-complex. Other studies have indeed focused on the use of a spatially-variable proportionality constant (see e.g. He et al 2014).

We clarified this in the revised manuscript (paragraph starting at line 70).

- Line 66-67: ".. one for temperature and another for net radiation". Doesn't this belong to the hybrid methods?

Yes, the reviewer is correct.

We listed this approach under hybrid methods (paragraph starting at line 51).

- Line 115 Formula (3) Is T the daily average temperature? Some formulations take into account the cumulated temperature which exceeds the threshold, measured for example with 1 hour time step. Would these different formulation affect the results?

Yes, T refers to the daily average temperature, as it is common practice in degree-day approaches. Considering the approach mentioned by the reviewer is an interesting alternative, which might produce a somewhat increased simulated snowmelt, since the daily temperature pattern might allow for snowmelt during some hours, even if the daily average temperature is below the threshold for snowmelt.

This, however, is beyond the scope of our study, since it is limited to simulations at a daily resolution.

- Section 2.2.1 Has the formula (5) been evaluated by using the available temperature data for the studied catchment?

As mentioned in the manuscript, this equation is derived from the analysis of observational temperature data throughout the year from a large number of stations situated at different elevations (Rolland, 2003). We did not evaluate this equation here again since the temperature driving data we use in this study were either from a gridded data product based on the interpolation of station measurements (Switzerland) or single station measurements (Czechia). Based on these data, it is not possible to properly evaluate the equation. We did, however, check a sample year from a Swiss catchment for which we obtained the lapse rate from the gridded data product and fitted a constant and sinusoidal lapse rate parameter (Figure 1).

[Figure]

Figure 1. Comparison between the temperature lapse rate as described by a constant and sinusoidal parameters and the observed values from a gridded temperature data product.

- Paragraph 2.2.5 What is the threshold used in the model as the maximum liquid water content retained in the pores (maximum water retention capacity)?

We set the maximum liquid water content retained in the pores as a free parameter for calibration and restricted the range between 0 and 0.2 following Seibert (1999).

- Section Results. I would plot the performance vs size of the catchment and altitude (also for a fixed configuration, given the high number of variable components).

The reviewer makes a good suggestion. Actually, we expected to observe some relationship between these parameters and model performance and we tested this. We even tested other parameters such as yearly snowmelt contribution to runoff (we mention it briefly in the manuscript, lines 362-364). Nevertheless, we did not find any clear relationships for our case study.

**References**

He, Z. H., Parajka, J., Tian, F. Q. and Blöschl, G. (2014). Estimating degree-day factors from MODIS for snowmelt runoff modeling. *Hydrol. Earth Syst. Sci., 18,* 4773-8789.

Rolland, C. (2003). Spatial and seasonal variations of air temperature lapse rates in Alpine regions. *Journal of climate*, *16*(7), 1032-1046.

Seibert, J. (1999). Regionalisation of parameters for a conceptual rainfall-runoff model. *Agricultural and forest meteorology*, *98*, 279-293.

**Authors' response to interactive comment by Reviewer #3 María José Polo**

We thank the reviewer for her valuable comments and suggestions to improve our contribution. Below we reply to each of them and explain how we will incorporate them into the manuscript.

This work analizes the performance of different snow routines based on the degree-day method in the framework of the HBV hydrological model. For this, runoff together with other snow-related variables are simulated in a large number of basins in Alpine areas in Central Europe and then compared to different sets of observations. The routines include different modifications for the snow routine components in HBV. De-spite the significant variability found among cases, the results identified an exponential snowmelt function as the best modification in terms of model performance, followed by the adoption of a seasonal degree-day factor; other processes, like refreezing, added little benefit to the model pointing out that complexity itself is not an advantage without careful model design. The work addresses an interesting topic for areas where physical modelling approaches demand larger data sets than the available observations, and it is very clearly presented. Despite the conclusions cannot be directly extrapolated to other snow regions in the world, the number of study cases cover a large area in Central Europe, where snow processes condition the hydrological response in many rivers. I have some observations that can be assessed by the Authors to emphasize the applicability of the results and the scope of the study; some minor comments are also included.

1. The work includes all the different snow routines in the HBV model, and no other hydrological model is assessed. I suggest making it clear in the title that the assessment is done on the HBV performance, since "...for runoff modelling in mountainous areas in Central Europe", since it may lead to expect a wider scope of models. Additionally, some comments addressing whether the level of improvement or not obtained from each routine is affected by the model choice. At least, some reference to similar models should be included and some justification of what conclusions would be expected to be shared from simulations by other hydrological models.

Indeed, this study is focused on the HBV model as all the analyses were done using this specific model. However, we think of this study as having a wider scope than HBV, in that we propose a methodology to evaluate the impact of using different model structures for a large array of catchments in hydrological models that use the degree-day method to simulate snow processes. In this respect, also the related comment by Juraj Parajka is interesting. Actually, he rather asked for interpreting our results more broadly beyond the relevance for just the HBV model.

He argued that this study might be interesting for other degree-day models, and asked to include some reference to the different implementations of this method in different hydrological models in the introduction. From this

260  perspective, HBV is just the tool to show and evaluate this methodology.

We expanded the introduction and discussion sections to clarify which aspects of our study are specific to the HBV model and which are of broader relevance for other hydrological models that use the degree-day approach.

2. A second issue is related to the spatial resolution of the input data, and potential scale effects. Gridded weather data in the Swiss cases, 1-km2 of gridded SWE, and25-m cell size of the DEM, whereas point observations from

265  stations and a 5-m DEM are used in the Czech catchments. Could you provide some assessment on these potential scale effects, and whether the source of weather data had an influence or not on the results? I also wonder whether using mean SWE values over each elevation zone, and point SWE measures, depending on the cases, could affect the results and comparison. Also, do you think that the results are scale-dependent of the cell size of the DEM used in the HBV model?

270  Regarding the DEM resolution, the cell size might have an impact on the results, but we argue that the proportions of the different elevation bands are represented correctly for both 5m and 25m resolution of the DEMs for most catchments in this study. This effect could become significant if the DEM would have a much coarser resolution (e.g. 500m) or if the catchments would be very small. In our case, we might only expect some minor effect in catchments with area less than about 10km2 (which are only two of the 54 selected catchments). The effect of,

275  for instance, the limited number of elevation bands (and the discontinuous and somewhat arbitrary choice of their elevation ranges) is probably much larger. Additionally, this factor may also be of importance for the snow model used to obtain the validation snow water equivalent data for the Swiss catchments, as topographical parameters such as slope and aspect need to be derived to correct for the influence of topography on snow distribution and redistribution.

280  Regarding the meteorological and SWE data, high-resolution data can become highly uncertain for individual points/grid cells, and these data should always be considered for somewhat larger areas. On the other hand, potentially high measurement errors and representativeness issues of the locality for the entire catchment/elevation band are also issues with observational data. We agree that the different approaches, i.e. catchment-wide aggregation of the gridded data product respective station data, might influence the results but

285  its impact is hard to quantify. That being said, we would expect that the model performance variability resulting from individual model structures would be similar.

We discussed these potential effects and their implications in the revised manuscript (paragraph starting at line 556).

3. In the introduction, I miss some inclusions, like the importance of sublimation from the snow under certain conditions (not only in dry areas like we reported in Sierra Nevada-Spain, but also during the summer in the Alps and other regions, see Herrero and Polo, 2016), the existence of experimental catchments in the world devoted to snow processes research (see for example a recent Special Issue in Earth System Science Data on "Hydrometeorological data from mountain and alpine research catchments"), or the use of remote sensing sources to provide data to monitor snow-packs and snowmelt (many examples can be found, e.g. Dietz et al. 2012). Lines 55-60 should also address the limitations of degree-day approaches, and when they, although simple, are not an option.

We thank the reviewer for pointing out these aspects that certainly will enrich the introduction and help to put this study into a broader context of snow hydrology. Nonetheless, we already had considered some of the suggestions by the reviewer but had at the time decided to leave them out to avoid the introduction becoming overly long. Other points, such as the limitations of the degree-day approaches (e.g. snow towers, page 3) were already included in the manuscript, but maybe not with enough emphasis.

We revised the introduction considering the suggestions by the reviewer (paragraph starting at line 51).

4. I am curious about the performance of each routine regarding the snow cover distribution. Did you check also their ability to capture this by testing against some satellite images? This is very interesting in terms of model performance to identify the sources of improvement or not.

We did consider using snow cover fraction as an evaluation metric for this study and performed some tests. However, in the end, we decided not to use it for different reasons. On one side, snow cover fraction does not provide a direct estimation of the amount of freshwater stored in the snow, which makes this parameter difficult to relate to the mass-balance approach of HBV. Additionally, cloud cover was an issue in the tests we performed. Besides, using this parameter could lead to large overestimations of snow water equivalent from, for instance, light snowfall events in late spring, when most of the catchment is no longer snow-covered but when there is still a significant storage of snow at high elevations, which would make the snow cover fraction jump up to 100% while the actual catchment-wide snow water equivalent would only have marginally increased. Finally, the scope of the study, including a large number of catchments and model alternatives, meant a large computational

315 demand. We, therefore, made an effort to identify the most relevant metrics for evaluating the model for both snow processes and rainfall-runoff transformation.

Considering additional metrics would certainly be very interesting and could add more value to the results but this is unfortunately beyond the scope of this study.

5. Since only four of the case studies were above 2000 m a.s.l. (only one above2500 m), I think that some comment
320 on how the results could change or not in higher elevation sites would shed light on their further applicability, especially in catchments where snowmelt is a higher fraction of runoff.

The reviewer raises an interesting question here. Indeed, only a handful of our catchments were at high elevations. There are few observations in high-elevation catchments and a lot of these catchments are influenced by glaciers. We took the decision to avoid glacierised catchments, as this would have required to increase the
325 model complexity, and therefore the complexity of the analysis. This decision limited the number of suitable high-elevation catchments. It is difficult to speculate about the applicability of these results for high-elevation catchments, as they tend to be small, with steep topography and large glacierised areas, scarcely vegetated, and more exposed to extreme weather conditions such as strong wind gusts. Additionally, the applicability of the results would also be limited by a general limitation of degree-day methods, which leads to the occurrence of
330 snow towers at high-elevations, where temperature hardly ever exceed the snowmelt threshold. We will include these considerations in the discussion.

We commented on this in the revised version of the manuscript (paragraph starting at line 547).

6. I fully agree with selecting just some examples to conduct the presentation of results. I think, however, that including more than just one catchment, and year, would add value to your results. You could suggest another
335 one from a lower altitudinal range, coming from the Swiss area, so that the impact of the spatial scale effects could, if needed, also be discussed. It would be very nice being able to see selected results from all the cases, I would suggest their inclusion as a supplement.

We agree with the reviewer in that including additional results, either more catchments or years, would improve the completeness of the manuscript and allow the reader to get more insights on the impacts of the different
340 model modifications. Nevertheless, we feel that even including an additional catchment or year, would imply overly-extending the manuscript with additional figures and make the whole presentation of the results more cumbersome. Nevertheless, if the editor agrees we could include an appendix with figures similar to Figure 4 (including validation results, following a later comment by the reviewer) for all catchments.

In an appendix to the manuscript, we included a table showing median efficiency values for each catchment, and objective function, both for model calibration and validation for both periods for the default HBV model. Additionally, we included a supplement containing further figures similar to Figure 4 (including validation results, following the previous comment), for all catchments.

Other comments:

7. The gridded data of SWE in the Swiss cases were derived from a temperature-index model. Could this bias the performance of the routines?

The temperature-index (TI) approach, in which the snow model we used to derive snow water equivalent is based on, includes a time-varying threshold temperature (Slater and Clark, 2006) to differentiate between snowfall and rain, and allows for mixed precipitation within a transition temperature range. Using topographical parameters such as slope and aspect, the model corrects for the influence of topography on snow distribution and redistribution. The model follows the parameterization proposed in Helbig et. al (2015) to derive fractional snow-covered area. Despite these features, using a TI approach for both the rainfall-runoff model as well as for the snow model providing validation data might indeed lead to some bias. Nevertheless, it has to be taken into account that the snow model makes use of a 3-dimensional sequential data assimilation (DA). The DA itself includes two methods which are based on spatially correlated error statistics. For snow accumulation, an optimal interpolation approach uses the snow water equivalent station data to correct the simulated snowfall amounts. Regarding snowmelt, an ensemble Kalman filter updates snowmelt rates as well as liquid water content. Finally, the combination of both data assimilation approaches results in corrections of modelled snow water equivalent within all 1 by 1 km grid cells. Magnusson et al. (2014) investigate the performance in predicting snow water equivalent when using this DA approach and compare it to the TI model without DA. Based on 1033 samples from 45 stations, they show that using DA leads to improved snow water equivalent predictions.

We included a brief discussion on the possible impact of using a degree-day model for both the hydrological model and the estimation of snow water equivalent for Swiss catchments in the methods section (paragraph starting at line 244).

8. Lines 259-260. Please, could you assess whether this decision could affect the results or not.

This decision might indeed have affected results, but the alternative would have caused a tremendous increase in parameter uncertainty and, thus, would have made the analyses almost impossible. In most of our catchments, elevation is the most important control on the spatial variation of snow processes, and this aspect is explicitly

considered by using the elevation bands (using somewhat wider/narrower bands would likely have minor impacts on results, see Uhlenbrook et al., 1999). The implicit consideration of different vegetation types in one vegetation zone is frequently used in catchment modelling to avoid over-parameterisation.

9. Figure 4. Please, could you show also some validation results for this example case and year.

We understand that having some validation results would allow the reader to better assess the model performance as well as the modifications presented in this study.

We modified Figure 4 to include validation results in addition to the calibration results.

10. Lines 410-412. Any comment on why these different behaviours are found?

Each individual modification of the snow routine adds between 1 and 2 additional parameters to the model. The design of HBV allows different parameters (even in different routines) to compensate for each other when calibrating the model. This issue is difficult to control for, especially when using automatic model calibration. Additionally, increasing the number of model parameters can also lead to over-parameterisation and equifinality issues. These different issues may lead to sub-optimal or physically inconsistent parameter sets that perform poorly when validating the model for an independent period. These potential issues lead us to be very careful in the model structures modifications we considered so as not to add too many additional parameters to the model.

11. Lines 425-427. Reading this, I would conclude that runoff data/simulations are somehow limiting the model performance's improvement (see also your comments in lines 482-484, and in lines 496-499). Additionally, this content should be reflected in conclusions (lines 565-567), to be more specific.

Good point. Yes, the evaluation against runoff data results in much smaller performance differences between the different model structures than the evaluation against snow water equivalent. This is to be expected as the ability of the model to simulate stream runoff is not only related to the structure of the snow routine but it is also affected by all other model routines that were not assessed in this study. We were aware of this issue but decided to perform the overall evaluation using these two objective functions based on two main considerations. First, if we want to evaluate changes on a particular routine of the model, we need to do it based on the output from the routine, not from the entire model, otherwise the noise from other routines of the model makes it impossible to attribute performance differences to any modification. That is why we used a metric based on snow water equivalent. Second, we were aware that HBV is not a perfect model and that it has issues with parameter compensation among others, and that the main application of the model is to simulate stream runoff. We, therefore, wanted to ensure that the modifications we introduced to the model were meaningful and produce

acceptable results despite of its imperfect nature. These two considerations were equally important to us and that is why we evaluated the evaluations in this way, even at the cost of obtaining relatively modest results.

We included this in the revised conclusions.

405  12. I would suggest including some quantitative result in the conclusions, but I leave it up to the Authors.

Quantitative results are related to our particular set of catchments and the choice of using the HBV model. In contrast, the broader implications of our study might be more challenging to express in quantitative terms.

We therefore decided not to include any quantitative results in the conclusions.

I hope that these comments help the Authors to address further their results and can contribute to the final

410  version of the manuscript.

We thank the reviewer again for the helpful comments that will certainly improve the quality of our manuscript.

**Authors' response to interactive comment by Reviewer #4 Thomas Skaugen**

430

We thank the reviewer for his valuable comments and suggestions to improve our contribution. Below we reply to each of them and explain how we will incorporate them into the manuscript.

**General comments**

This paper describes the testing of many alternative conceptual algorithms for snow modelling implemented in
435  the Swedish HBV model. The suitability of the different algorithms has been assessed by split sample procedures
for many catchments in Czechia and Switzerland. The paper is well written, well organized and the experimental
setup seems, in principle, to be fine. However, the possible improvements of the tested alternative algorithms
are extremely subtle and the authors recommend exponential snowmelt function and seasonally varying degree-
day factor based on tiny improvements which have not, as far as I can see, been tested for their significance. I
440  think the objective of the paper is good, it would be nice if we in objective ways could agree on improved concepts
in snow modelling that when implemented would improve any model, but I am doubtful if the current methods
are up for the task. The following issues need to be addressed in order to make paper suitable for publication.

Yes, the effects of the different modifications are small. Nevertheless, even if the average model performance
improvement for the recommended model modifications are small on average, these modifications are significant
445  for individual catchments, and do not lead to decreased model performance in any case. Minor changes had to
be expected in general because we use a large sample of catchments in our study. Any improvements will, thus,
tend to average out and look less impressive. We argue that while these improvements might indeed be small,
our evaluation based on many catchments means that the findings are more robust than in many previous
studies.

450  1) The paper misses a major investigation on equifinality issues (see papers of K. Beven and J. Kirchner on this
topic). The HBV model itself has a lot of freedom, i.e. parameters to be calibrated, and most of the suggested
algorithms for possible improved snow modelling add calibration parameters and hence to the problem of
overparameterization. The point is that many of the suggested snow model modifications may have potential for
being better at modelling snow, but the effect is impossible to isolate due to the
455  overparameterization/equifinality. I have personal experience with trying to implement, what I thought was
brilliant, ideas of improved snow modelling to the HBV model. They were all insignificant, and after a while I

realized that the compensating powers of all the parameters in HBV made it impossible to isolate and assess the effect of new algorithms (the frustration inspired the development of a new rainfall runoff model). The inclusion of the objective function for SWE is a step in the right direction, it narrows the freedom of the parameters, but probably not enough (you could try to also include Snow Covered Area, SCA). How many calibration parameters are there in the various model configurations? Are the numbers acceptable by any measure? Are their ranges physical at equifinality?

Indeed, equifinality is an issue in many hydrological models, and HBV is no exception to this. However, compared to many other models, the HBV model uses rather few parameters and parameter uncertainty is thus, smaller. The particular version used here, HBV-light, has been frequently used to address parameter uncertainty in the past years. So, while parameter uncertainty is an issue, we argue that we in the past have gathered quite some experience related to this issue. That being said, and while we did not include this in the manuscript, we performed a Monte Carlo sensitivity analysis on the HBV parameters. We found that, even if some of the variants (i.e. $T_{p,m}$, $\Delta P_e$, or $C_{0,s}$) produce compensating effects on some parameters (e.g. PCALT, FC, LP or BETA), this effect was only observed for some of the catchments. Overall, parameter values and sensitivity tended to be fairly consistent across all the tested model variants for most of the catchments. As we previously mentioned, we took consideration of both potential equifinality and parameter compensation issues in our analysis.

We emphasised this in the revised manuscript. Furthermore, we mentioned and discussed the sensitivity analysis both in the methods, results, and discussion sections (paragraphs starting at lines 329, 508, and 572) but we did not include any further figures in order to not make the manuscript even more complex.

Most of the modifications that we tested in this study add only one extra parameter to the snow routine of the model (which consists of 5 parameters: degree-day factor, refreezing coefficient, threshold temperature, water holding capacity of the snowpack, and snowfall correction factor). Additionally, as the reviewer noticed, we assessed the impact of these modifications on the output of the snow routine (i.e., snow water equivalent) to avoid interactions from the other model routines and parameters. We also tested the use of other snow-related objective functions such as snow cover fraction. However, in the end, we decided not to use this measure because snow cover fraction does not provide a direct estimation of the amount of freshwater stored in the snow, which makes this parameter difficult to relate to the mass-balance approach of HBV. Additionally, cloud cover was an issue in the tests we performed. Furthermore, using this objective function could lead to large overestimations of snow water equivalent from, for instance, light snowfall events in late spring, when most of the catchment is

no longer snow-covered but when there is still a significant storage of snow at high elevations. Such events could make the snow cover fraction jump up to 100% while the actual catchment-wide snow water equivalent would only have marginally increased. Finally, the scope of the study, including a large number of catchments and model alternatives, meant a large computational demand. We, therefore, made an effort to identify the most relevant

490 metrics for evaluating the model for both snow processes (i.e. Rw) and rainfall-runoff transformation (i.e. Rln(Q)). Regarding the decision to also evaluate the results respect to stream runoff, it was taken based on the common use of many hydrological models. We know that HBV is an imperfect hydrological model (as is the case for all models) based on certain assumptions that lead to issues such as parameter compensation. Even so, since models based on these assumptions will continue to be used in the foreseeable future mostly for runoff simulation, we

495 wanted to ensure that the modifications we introduced to the model would produce acceptable results within the imperfect framework of the model. We agree with the reviewer in that better modelling approaches need to be found, but we also think that the available tools need to be evaluated and improved upon as well.

In the revised manuscript we clarified the choice of objective functions to perform the evaluation (paragraph starting at line 282).

500 Coming back to the number of additional calibration parameters for the various model configurations, this number varies between 1 and 3 parameters for single modifications. Several of the modifications that we selected were derived from observations, such as the seasonally-variable temperature lapse rate (Rolland, 2002) or the exponential precipitation phase partition (Magnusson, 2014). For the other parameters, we used constrained ranges (based on, for instance, Seibert (1999) for the default HBV structure) to ensure that parameter values do

505 not become unrealistic. When different modifications are used simultaneously, the number may go up to 9 parameters. We regard this number as excessive and a clear over-parameterisation, which opposes the aim of preserving a simple structure with as few parameters as possible. However, we still included this variant in the evaluation for the sake of completeness.

We clarified this in the revised manuscript (paragraphs starting at line 329, 426, and 584).

510 2) I would desire a more stringent terminology. Words like "efficient" and "complex" have really lost their true meaning in the literature of hydrological modelling. Effective parameters really mean parameters that lump many processes or represents areal averages and has little to do with efficiency. A non-linear formulation of a process is not necessarily complex if the parameters are physical and measurable. To me, an over-parameterized model where, due to the compensating behavior among the parameters, the degree-day factor is suddenly correlated

515    to the parameter controlling the subsurface storage capacity is infinitely complex. Please consider rewriting the paragraph that starts at 525

We understand the concerns of the reviewer and agree with the need for a clear and concise terminology. Ideally, model efficiency relates to the definition the reviewer provides here. Nevertheless, as the reviewer also mentions, these terms can also be used to refer to other concepts such as models that provide acceptable results, even if

520    for the wrong reasons. As explained in reply to the previous comment, in this study, we aim for both improving the quality of the processes conceptualisation in the model and ensuring that this improved conceptualisation works well with the imperfect nature of the model.

We revised the manuscript to ensure that the terminology we used was appropriate and concise. Additionally, rewrote the aforementioned paragraph to iron out inaccurate references to complexity and efficiency.

525    3) There are several paragraphs subjectively praising the HBV model for its ability to simulate hydrological behavior for various catchment types (p. 24, l.476, p.25, l 520-25, p.26, l563). "Hydrology" is a wide term and comprises more than runoff (and SWE admittedly), what about the subsurface, SCA, evapotranspiration etc. How come we are just presented result for one catchment?

The reviewer makes an important remark here to the need for objectively assessing the strengths and weaknesses

530    of the selected models and design choices.

We revised the passages mentioned by the reviewer to ensure that the demands for objectivity were met adequately.

Regarding the term "hydrology", we agree with the reviewer in that hydrology is a wide term and that there are many relevant variables and processes that often get overlooked in favour of – in most cases – stream runoff. We

535    refer to the reply to major comment 1 for an explanation on the variables we used to assess the analysis presented in this contribution.

Similar to the previous point, we considered this comment when revising the manuscript and we replaced the generic references to "hydrology" by more concise terms.

We only presented results for a single catchment to not make the paper excessively long or complex. The analysis

540    we present here includes a large array of catchments and alternative model variants, which make it impractical to present all the results in a detailed way without making the manuscript overly cumbersome.

In an appendix to the manuscript, we included a table showing median efficiency values for each catchment, and objective function, both for model calibration and validation for both periods for the default HBV model.

Additionally, we included a supplement containing further figures similar to Figure 4 (including validation results, following the previous comment), for all catchments.

**Specific comments**

P1, l.18, "popular" subjective

We agree with the reviewer that this is a subjective expression.

We rephrased the text to emphasise that this model has been (and still is) widely used in many different settings.

P1, l.27 "optimal degree of realism", rephrase

We rephrased the expression based on the responses to the major comments above.

P2, l.143-44 How can "the limitations of data availability–" "pose a challenge to properly monitoring". Rephrase....

With this sentence we wanted to express that monitoring hydrological processes (and more specifically snow processes in this case) is challenging with limited observations.

We rephrased the sentence for clarification.

P2, l.45-46 "Furthermore..." This sentence does not relate to anything above.

The idea we wanted to express with the final part of this paragraph was that, if having (limited) observations already makes it challenging to properly assess the current evolution of the hydrological processes (in this case snow processes), to be able to predict their future evolution with – obviously – no observations on the potential changes is even more complicated.

We rephrased this sentence to make it fit better in the paragraph.

P2, l.52 ..available at..

We corrected this mistake.

P2, l.53. ..in a distributed way.. Not always, see Skaugen et al., 2018 (Hydrology Research)

We admit that we over-generalised the identification of energy-based approaches with distributed hydrological models. We thank the reviewer for providing this reference.

We addressed this by explicitly mentioning lumped and semi-distributed models that are based on these approaches (paragraph starting at line 51).

P2, l.58 ..relevant for.. Aren't they relevant everywhere?

Indeed, using radiation data in addition to temperature data is relevant everywhere. Nevertheless, the benefits of using these approaches are most notable in the catchments described in the aforementioned sentence than in other types of catchments, where the impact is more modest. We did test such an approach for this set of

catchments and found that the improvements were small while requiring an additional data source and calibration parameter.

575 P2, l.62 ..distribution function.. What is this?

We intended to say "bucket-type model" / "conceptual model".

We corrected this error in the revision.

P3, l.83-84. ..and investigated whether.. See major comment above.

We refer to the reply to major comment 1 and 2 above.

580 We revised the text to ensure that the terminology we used is concise and relevant.

P4, l.98. "well established", what does this mean? is it good or just old

We would argue that it means a bit of both. Indeed, the degree-day approach is an old conceptualisation of snowmelt processes, which has been in use for a long time already, and it has been evaluated, tested, and implemented in many different studies and models due to its low data requirements and explanatory power.

585 We rephrased this expression to include these nuances.

P5, l.125...to it... Refers to HBV or the individual components

It refers to the snow routine of HBV.

We reformulated this to avoid confusions.

P5, l.125 is precipitation lapse rate missing in the table? Could we have all calibration parameters in the table?

590 We intended this table to present the proposed modifications to the snow routine of HBV so, since we decided not to test any alternative to the precipitation lapse-rate, we did not include it in this table.

We added this component to the table.

Regarding the calibration parameters, we argue that including all calibration parameters is out of the scope of this table.

595 We added a sentence in Section 2.1 (in which the individual parameters are described) summarising the number of calibration parameters in the snow routine of the model. The reader will then be able to easily calculate the number of calibration parameters that are needed for each model variant.

P5, l.131. Heading, "Temperature and precipitation lapse rates"

We modified the section header (see also the previous comment).

600 P7, l.189. ..if somewhat.. How is it more realistic

It is more realistic than the one used in HBV since it does not have an abrupt transition (i.e. snowmelt being 0 up to a threshold and increasing linearly thereafter) in the change of snowmelt rate. Nevertheless, it does require the use of an additional parameter to control for the smoothness of the snowmelt transition.

P12, l.257. ..higher model complexity.. why more complex if you increase the temporal resolution

We explain this in the following sentences. We argue that to represent these processes at a sub-diurnal time step correctly, we would need to include additional parameters to control for processes that become relevant at these resolutions. Nevertheless, "complex" might not be the appropriate term here; "detailed" might be more suitable in this context (see also the reply to major comment 2 above).

We rephrased this sentence to be more concise in the terminology.

P12, l.257-58. "Other factors..", please elaborate

With this sentence, we meant that factors such as the transport time of meltwater from the snowpack to the stream become relevant for sub-daily time steps.

P12, l.266-67. Good, this fights the problem of over-parameterization. Could even include SCA

We refer to the reply to major comment 1 for a discussion on why we finally did not use snow cover fraction as a metric to evaluate this study.

P13, l.292. "efficiency", efficient how? Faster? gets more done? or just better?

In here, we referred to model performance when evaluating the model against each of the chosen objective functions.

As already mentioned in major comment 2 above, we revised the manuscript to ensure that the terminology is concise and relevant throughout the text.

P14, l.322..performance for...

We corrected the error.

P14, l.323-24. Can we accept improved performance for snow and decreased performance for runoff? I know that other authors have reported this, but is this not a clear indication of model structure flaw? Please elaborate, this is important

The reviewer raises an important point here, which is linked to the major comment above. Indeed, this kind of observations should make the hydrological community think about the complexity issues and limitations of the current generation of models and use this evidence to guide further research efforts that allow us to increase our

understanding of these processes (including their connexions and feedback mechanisms) as well as to design and implement better (and usable) modelling strategies that avoid these issues.

Nevertheless, this issue is beyond the scope of this manuscript, which attempts to improve an existing, yet imperfect tool by exploring, testing, and evaluating the suitability of existing alternative structures.

P18, l.373.. "catchment dependent.." I do not understand this sentence

With this sentence, we wanted to express that there are some modifications which have a clear and consistent impact on most catchments (either negative or positive), while other modifications produce either positive or negative impacts depending on the catchment.

We rephrased the sentence to make it easier to understand.

P19, Figure. What does the y-axis represent, I struggle with this figure

The Y-axis shows the rank spread of each modification across all the catchments in the study, and each column adds up to 100%. So, for a given modification it shows for which percentage of catchments it is the best model structure, the second-best model structure, and so on. So, for instance, for the top left subplot, using a seasonal degree-day factor is the best alternative (among all the single modifications + default HBV structure) for ~80% of the tested catchments, the second-best alternative for ~10% of the catchments, and so on.

We clarified this in the text.

P20, l.404-407. This paragraph is very complex, can you please explain better

This paragraph is intended to clarify why, in the case of introducing 5 modifications to the snow routine (i.e. modifying each of the snow routine components that we evaluated), there are only three possible alternatives. The only available alternative representations (see also Table 1) are used for the lapse rate (i.e. $\Gamma_s$), the threshold temperature (i.e. $T_{P,M}$), the degree-day factor (i.e. $C_{0,s}$), and snowmelt and refreezing (i.e. $M_e$), in combination with one of the three alternative representations for the precipitation phase partition (i.e. $\Delta P_l$, $\Delta P_s$ or $\Delta P_e$).

We decided to remove this paragraph to avoid confusions.

P20, l.408 64 or 63 (see Table above)

We considered 64 different model structures, including the default HBV structure. In Table 3 we only showed modifications to the default HBV structure. Nevertheless, we understand that this can lead to confusions. We changed the table accordingly.

P23, l.457 ..are dominant.. meaning strong or better?

In here, by "dominant" we meant the modifications to single components that appear most frequently in the top-ranking model variants.

We rephrased this passage to make it more concise.

660    P24, l.475. The first sentence is meaningless. Of course it is difficult to improve hydrological models, the processes are complex. The reason why it is difficult in the case of HBV could be due to the over-parameterization, not because it has been widely used with acceptable results.

This sentence was meant as an introduction to the discussion so, taking into account the major comments by the reviewer, we will modify just to state that we observed that it is difficult to improve hydrological models like HBV.

665    We included some comments in the discussion section on why such models are difficult to improve.

P24, l.487. .. runoff is modulated.. Rephrase

The intention with this sentence was to stress that, as already pointed out before, the final model output is the result of the interaction between the different routines of the model. This, as the reviewer points out in another comment, may be related to compensating effects between parameters but also to a loss of the signal from any

670    modifications made on the snow routine. It is, therefore, to be expected that efficiency changes are minor when evaluating the model based on this variable.

We rephrased the sentence taking the previous discussion into account.

P25, l.533...even if model complexity...in a sensible way.. The sentence is strange

The intention with this sentence was to point out that, if there were enough data available and knowledge about

675    the processes that to be simulated, then it would be justified to add more complexity (here understood as a more detailed description of the processes) to the model.

We rephrased the sentence to make it clearer.

P26, l.551 different settings.. please be more specific.

By different settings, we were referring to the geological, geographic, climatological, and hydrological

680    characteristics that define the hydrological behaviour of a given catchment.

We made this sentence more specific.

P26, l.563. Unsubstantiated, we have only seen the result for one catchment.

We understand the concern of the reviewer regarding drawing general conclusions about the whole study when we only presented the details of a single catchment in the manuscript. We took this decision to facilitate the

685    storyline of the manuscript and not overwhelm the reader with endless results.

We refer to major comment #3 for the relevant modifications concerning this comment that we included in the revised manuscript. We hope that the aforementioned changes provide enough evidence to support this conclusion.

P26, l.565. How to proceed with this "better approach", how to do it in practice?

690 In this conclusion point, we state that carefully assessing which objective, the necessary level of detail (see previous comment on complexity vs detail), and data availability to each case is a better approach than just picking whichever model we are familiar with or have a preference for. Obviously, this is easier said than done but, in this contribution, we aim to provide a methodology to do such an assessment over a large sample of catchments and model structure variants for a specific purpose.

695 We clarified this point in the revised manuscript.

[revised manuscript text omitted]

---

## Author Response (AR2)

Dear HESS Topical Editor Elena Toth,

Thank you again for your efforts with our manuscript. We have now included the minor clarification requested by reviewer María José Polo. In addition, we have corrected a small mistake in the caption of Figure 4 (i.e. wrong dates).

We look forward to continued advancements in this very interesting topic.

Best regards,

Marc Girons Lopez, on behalf of all authors

[revised manuscript text omitted]